# Electrophysiological priming effects demonstrate independence and overlap of visual regularity representations in the extrastriate cortex

Alexis D. J. Makin[1]*, John Tyson-Carr[1], Yiovanna Derpsch[1], Giulia Rampone[1], Marco Bertamini[1,2]

**1** Department of Psychological Sciences, University of Liverpool, Liverpool, United Kingdom, **2** Department of General Psychology, University of Padova, Padova, Italy

* Alexis.makin@liverpool.ac.uk

## Abstract

An Event Related Potential (ERP) component called the Sustained Posterior Negativity (SPN) is generated by regular visual patterns (e.g. vertical reflectional symmetry, horizontal reflectional symmetry or rotational symmetry). Behavioural studies suggest symmetry becomes increasingly salient when the exemplars update rapidly. In line with this, Experiment 1 (N = 48) found that SPN amplitude increased when three different reflectional symmetry patterns were presented sequentially. We call this effect 'SPN priming'. We then exploited SPN priming to investigate independence of different symmetry representations. SPN priming did not survive changes in retinal location (Experiment 2, N = 48) or non-orthogonal changes in axis orientation (Experiment 3, N = 48). However, SPN priming transferred between vertical and horizontal axis orientations (Experiment 4, N = 48) and between reflectional and rotational symmetry (Experiment 5, N = 48). SPN priming is interesting in itself, and a useful new method for identifying functional boundaries of the symmetry response. We conclude that visual regularities at different retinal locations are coded independently. However, there is some overlap between different regularities presented at the same retinal location.

## Introduction

Electroencephalography (EEG) studies have identified a symmetry-related ERP called the *Sustained Posterior Negativity* (SPN, for review see [1]). Amplitude is more negative at posterior electrodes for symmetrical than asymmetrical stimuli [2–4]. The SPN is a difference wave that begins immediately after the visual N1 (at around 200–250 ms post stimulus onset). Larger SPNs are those where amplitude much more negative in symmetrical than random conditions.

The SPN has been well characterized: Amplitude scales with the salience of different symmetry types [5] and with the proportion of symmetry in symmetry plus noise displays [6]. An

**Data Availability Statement:** All stimuli and image construction algorithms are available on Open Science Framework (OSF), along with ERP analysis

materials and Supplementary Materials 1 and 2 (https://osf.io/2yjus/).

**Funding:** This project was part funded by an ESRC grant award to Marco Bertamini (ES/K000187/1) and part funded by an ESRC grant awarded to Alexis Makin (ES/S014691/1).

**Competing interests:** The authors have declared that no competing interests exist.

example of the parametric SPN response to symmetry is shown in Fig 1. The SPN is found whether participants are actively discriminating regularity, or performing secondary tasks [7]. Source localization suggests that the SPN is generated by the extrastriate visual cortex [3,8]. This is consistent with functional Magnetic Resonance Imaging (fMRI), which has identified a parametric response to symmetry in a network of extrastriate areas including V3, V3a, V4, VO1 and LOC, but not in the primary visual cortex [8–12]. Throughout this paper we refer to 'the extrastriate symmetry network' that 'generates the SPN', while of course acknowledging that the same posterior brain regions mediate many other visual computations.

Behavioural studies have also identified systematic influences on symmetry perception [13]. One such influence was reported by Sharman and Gheorghiu [14], who found that symmetry in rapidly changing displays is discriminated more easily than symmetry in static displays. This dynamic symmetry advantage was also found by Niimi, Watanbe and Yokosawa [15].

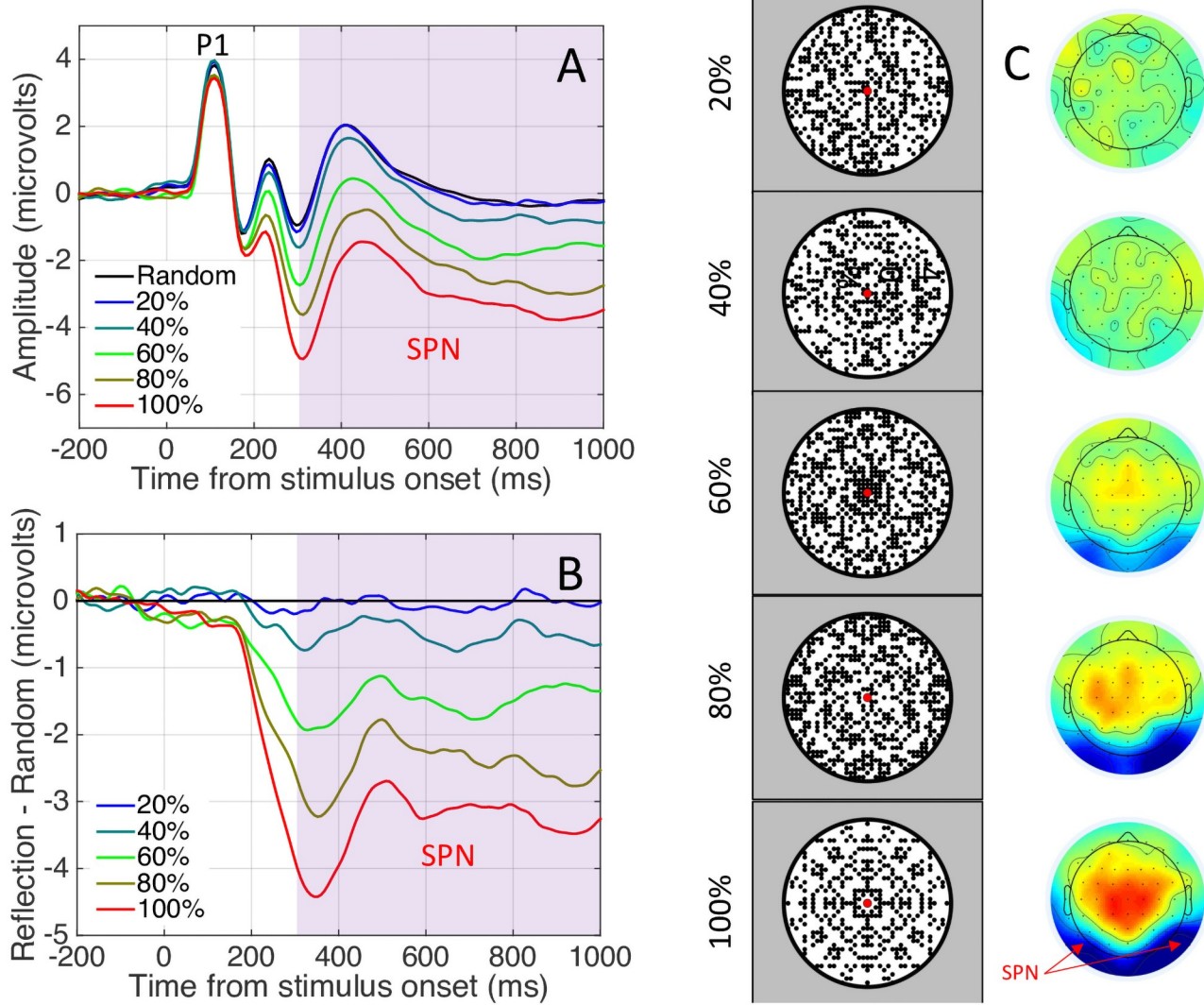

**Fig 1. Example Sustained Posterior Negativity from Palumbo et al. [6].** A) Grand average ERP waves from posterior electrodes. P1 and the SPN intervals are labelled. B) SPN as a difference wave (reflection–random). Larger SPNs are ones where this difference wave falls a long way below zero. Here SPN amplitude scaled with the proportion of symmetry in the image. C) Example stimuli with 20 to 100% symmetry and corresponding topographic difference maps. Here the SPN appears as increasing blue at posterior electrodes. In the current series of experiments, we investigated how the SPN changes with repeated presentation.

Given these results, Sharman and Gheorghiu [14] concluded that *"It might be the case that continuous rapid presentation of symmetrical patterns, could result in an SPN that increases in amplitude over time." (page 8).*

Our Experiment 1 explored SPN repetition effects and confirmed Sharman and Gheorghiu's [14] prediction. We term the increase in SPN amplitude 'SPN priming'. Experiments 2–5 then exploited this SPN priming to determine functional boundaries within the extrastriate symmetry network. Repetition effects are often used to determine functional boundaries in this way: If prior presentation of stimulus A has no effect on the response to stimulus B, then A and B are coded independently. Conversely, if prior presentation of stimulus A alters the neural response to stimulus B, then there is some overlap between A and B in the brain. Transfer of repetition effects indicate representational overlap, while absence of transfer indicates representational independence [16–21]. We thus tested whether SPN repetition effects survive changes in location (Experiment 2), orientation (Experiments 3 and 4) and regularity type (Experiment 5).

## Experiment 1. Repetition effects with different and identical reflections

The trial structure of Experiment 1 is shown in Fig 2A. Each trial began with a 1500 ms blank period. This was followed by a sequence of three abstract black and white patterns (500 ms each, separated by 200 ms gaps). The three patterns were either random or regular with salient 8-fold reflectional symmetry (Fig 2B). Participants identified oddball trials with a grey blank disk at the second sequence position. We used this secondary task because we were interested in the dynamics of spontaneous regularity coding, rather than repetition effects on cognitive classification.

As mentioned, the SPN is a difference wave. We expected amplitude to be lower for reflection than random patterns. When refer to SPN priming, we mean the difference between reflection and random waves increases across the three successive presentations.

Experiment 1 explored SPN repetition effects with identical and different exemplars (Fig 2B). Sharman and Gheorghiu's results predict SPN priming in the different exemplars condition only. Experiment 1 also tested the specificity of SPN priming. We contrasted simple and complex patterns, which differed in spatial frequency content and other low-level visual features (see left and right columns of Fig 2B). This was a between-subjects factor, with 24 participants each condition. Finally, it is likely that mere repetition of any stimulus elicits some non-specific effects. Therefore, we also included a condition where a reflection pattern is presented third, following two randoms (RandRandRef condition). The SPN response to this third reflection allows us to estimate this non-specific repetition effect.

### Experiment 1 methods

General methods for all experiments are described at the end of the manuscript. We provide a few essential details here. Experiment 1 had 48 participants (age 18–43, mean age 20, 9 males, 4 left-handed). Each trial began with a 1500 ms blank with white placeholder disk and red fixation dot in the centre. On most trials, three patterns were then shown for 500 ms, separated by 200 ms intervals. On rare oddball trials, the second element of the sequence was a blank grey disk. On each trial, participants responded 'all patterns' or 'blank in the middle', using the A and L keys to enter their responses (Fig 2A). The response key mapping switched unpredictably (A for blank in the middle, L for all patterns, or vice versa). There were an equal number of trials with each key mapping.

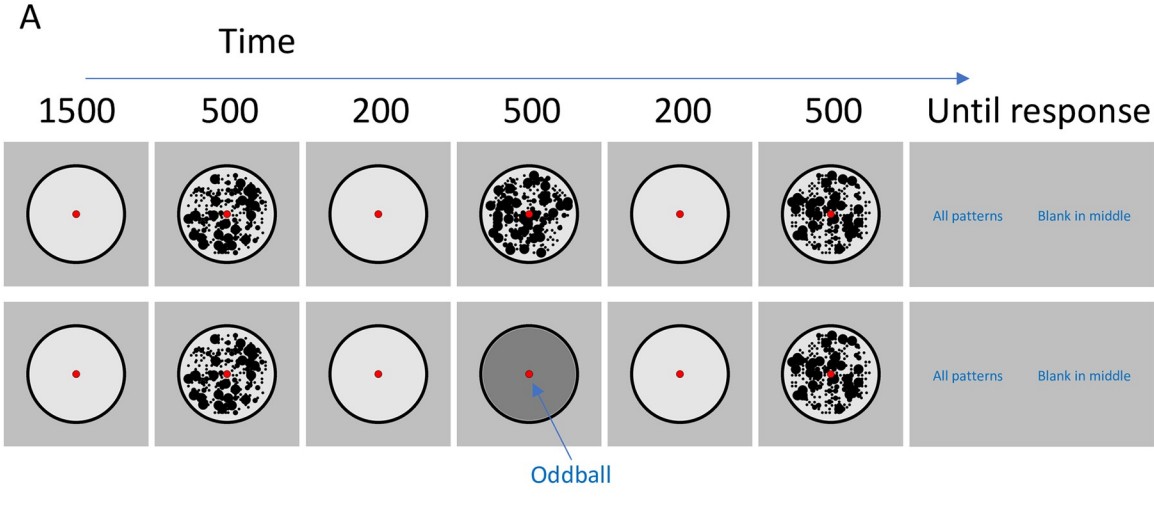

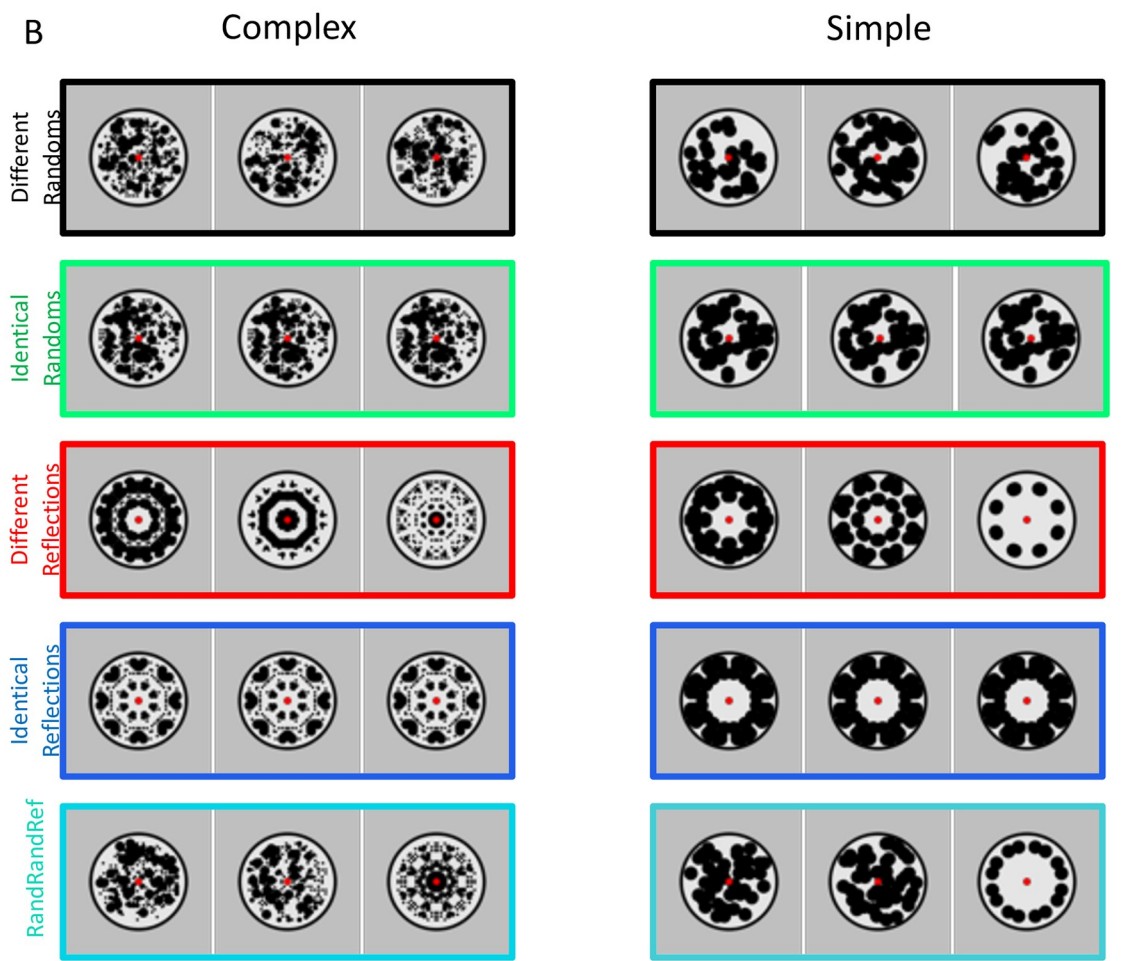

**Fig 2. Experiment 1 method and stimuli. A)** Trial structure in normal and oddball trials. All trials involved a sequence of three 500 ms presentations, separated by 200 ms gaps. Participants responded by pressing buttons for "All patterns" (all three presentations involved black and white patterns, as in the top row) or "Blank in the middle" (the second item in the sequence was blank, as in the lower row). **B)** Different triplets used. 24 participants were presented with complex patterns (left column) and another 24 were presented with simple patterns (right column). Individual patterns shown here are examples, in the real experiment each trial used different patterns. The coloured frames are matched with the colour of ERP waves in subsequent figures.

The experiment was divided in 15 blocks of 36 trials (540 trials in total). Participants had a break in between each block, and the electrodes were checked if necessary. Trials in each block were presented in random order. In the main experiment there were 240 different random trials (where all three patterns had a random arrangement), and 60 trials from each of the other conditions (Fig 2B). There were also 60 additional oddball trials (11.1%) requiring a *blank in the middle* response. These oddball trials were not included in EEG analysis. The proportion of triplet types in each block was identical and thus matched the whole experiment. An additional single block was presented as practice (not included in EEG analysis). Participants gave the correct answer on 97% of trials. Trials were included in EEG analysis even if participants gave the wrong behavioural response. Given the moderate cognitive demands of the unpredictable key mapping, it is likely that most errors were introduced at the response entry stage. Most errors are unlikely to reflect genuine confusion about whether there was a blank in the middle.

All stimuli and image construction algorithms are available on Open Science Framework (OSF), along with ERP analysis materials and Supplementary Materials 1, 2 and 3 (https://osf.io/2yjus/).

## Experiment 1 results

Fig 3A shows grand-average ERP data from the posterior electrode cluster [PO7, O1, O2 and PO8], averaged over the between-subjects factor Complexity (simple, complex). Unsurprisingly,

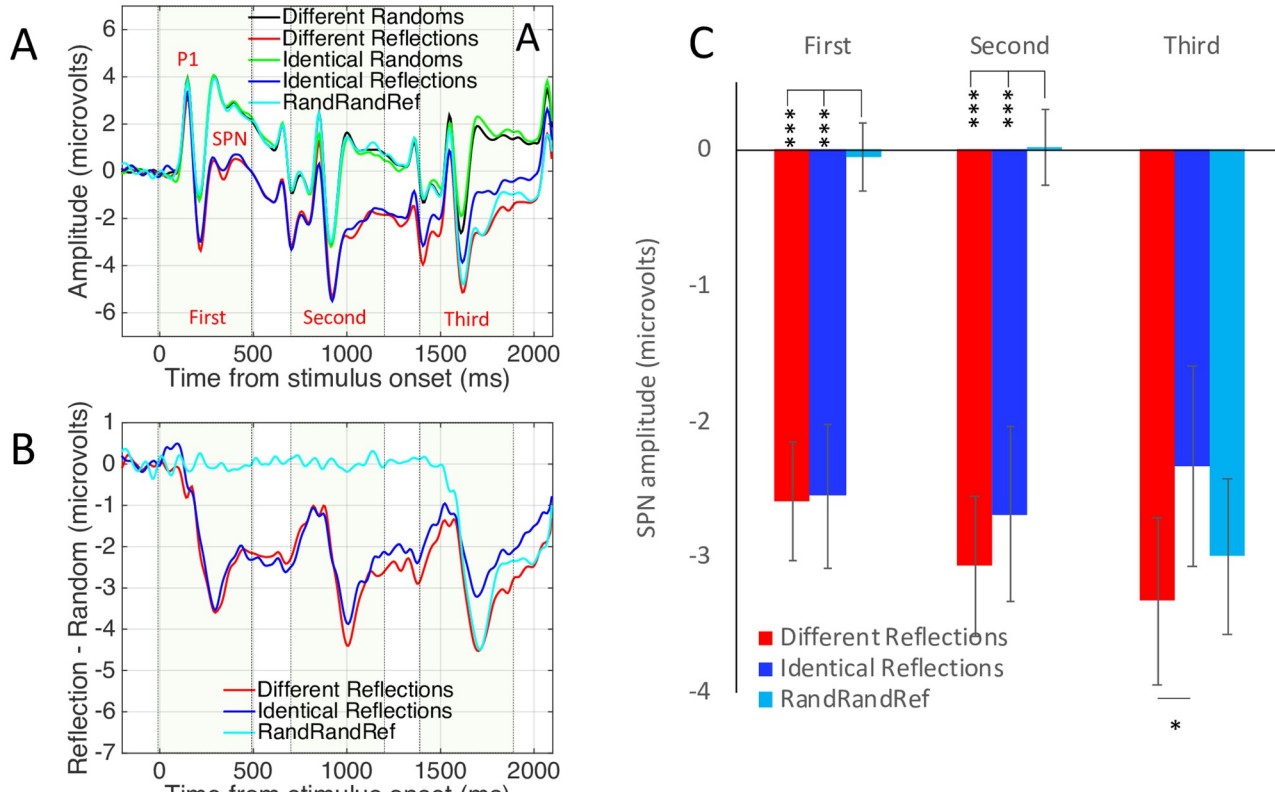

**Fig 3. Experiment 1 results. A)** Grand Average ERPs from electrode cluster O1, PO7, O2 and PO8. Colour corresponds to coloured outlines in Fig 1. Results are averaged over the Complex and Simple patterns. **B)** The SPN shown as a difference wave. The intervals where the patterns were visible on the screen are shown with dotted boxes. **C)** Mean SPN for first, second and third intervals. Note SPN priming in the different reflections condition (red), but not in the identical reflections condition (dark blue). Results from Experiment 1. Error bars = 95% confidence intervals (if they do not cross zero, the amplitude difference from zero significant at the .05 level). Stars show significant pairwise comparisons within an interval (* p <0.05, *** p <0.001).

there are three Visual Evoked Potentials (VEPs), driven by the three successive onsets. The waves for different and identical random sequences were very similar (green and black waves in Fig 3A). The waves for different and identical reflection sequences diverged over the trial (red and blue waves in Fig 3A).

The SPN difference wave also had three peaks, although amplitude remained below zero throughout (Fig 3B). When three different reflections were presented, SPN amplitude increased with repetition (red trace). This repetition enhancement effect is referred to as S*PN priming*. When three identical patterns were presented, there was no SPN priming (blue trace). In the RandRandRef condition, the third pattern (a reflection) generated an SPN of intermediate amplitude (light blue trace).

The nature of SPN priming requires clarification: We did not observe three discrete SPNs. Instead, this should be interpreted a long difference wave, with a single 200 ms pre-stimulus baseline. The long difference wave selectively enhanced by a sequence of three of changing exemplars, but not identical exemplars.

To examine this selective SPN priming effect statistically, the difference from corresponding random wave was measured in the final 250ms of each presentation interval and 100 ms into the ISI (First window = 250–600 ms, Second window = 950 to 1300 ms, Third window = 1650 to 2000 ms). Average SPN amplitude from these time windows is shown in Fig 3C. Bars in Fig 3C represent mean amplitudes significantly lower than zero (all one sample t tests significant $p < 0.001$, see 95% confidence intervals).

Effects were analysed with a mixed ANOVA. There were two within-subject factors [3 Sequence position (first, second, third) X 3 Sequence type (different reflections, identical reflections, RandRandRef)] and one between-subjects factor [2 Pattern complexity (complex, simple)]. Here we only report the most theoretically interesting effects for the sake of brevity. The full results of this ANOVA analysis are reported in Supplementary materials 3. Note that the Greenhouse Geisser-correction factor was used to adjust degrees of freedom when the assumption of sphericity was violated, so some non-integer DFs are reported. There were main effects of Sequence position (F $(2,92)$ = 56.179, $p < 0.001$, $\eta_p^2$ = 0.550) and Sequence type (F $(2,92)$ = 37.906, $p < 0.001$, $\eta_p^2$ = 0.450) and an interaction between Sequence position and Sequence type (F $(2.752, 126.594)$ = 54.052, $p < 0.001$, $\eta_p^2$ = 0.540). There was no effect of the between-subjects factor Complexity (F $(1,46)$ = 3.627, $p = 0.063$) and no interactions involving Complexity ($p > 0.116$).

The three-way interaction was largely driven by the unique nature of the RandRandRef sequence (reported in Supplementary materials 3). However, the most theoretically interesting comparison between the Identical and Different Reflections sequences. We thus ran a 2X3 follow up ANOVA to compare these conditions [Sequence position (first, second, third) X Sequence type (different reflections, identical reflections)]. There was a significant interaction (F $(1.520, 71.437)$ = 7.945, $p = 0.002$, $\eta_p^2$ = 0.145). This was due to a significant linear effect of Sequence position in the different reflections condition (F $(1,47)$ = 13.674, $p = 0.001$, $\eta_p^2$ = 0.225) but not in the identical reflections condition (F $< 1$).

Next, we analysed amplitude of the third SPN with one factor repeated measures ANOVA [3 Previous patterns (different reflections, identical reflections, randoms)]. There was a significant influence of Previous patterns (F $(2,94)$ = 4.570, $p = 0.013$, $\eta_p^2$ = 0.089). The SPN was larger after two different reflection exemplars than two identical reflection exemplars (t $(47)$ = 2.657, $p = 0.011$). The third pattern from the RandRandRef sequence (e.g. the reflection) generated an SPN of intermediate amplitude, which did not significantly differ from the other conditions (different reflections t $(47)$ = -1.136, $p = 0.262$; identical reflections sequence t $(47)$ = 1.980, $p = 0.054$). This aspect of the results was somewhat inconclusive (t-tests have different DF to ANOVA because the between subject's factor Complexity is not included).

This basic ERP analysis was based on a subset of electrodes and time windows. Rather than running whole scalp analysis with Electrode cluster as an additional factor, we complemented the above with mass univariate (LIMO) and global field power (GFP) analysis, which incorporates all electrodes and time points. This is a more efficient way of imaging the full data set (see Supplementary materials 1).

## Cortical sources of the SPN priming effect

We interpreted the SPN priming effect as an increase in amplitude of the bilateral extrastriate symmetry response. However, an alternative explanation is that SPN priming is caused by additional activations emerging elsewhere—ERPs from different cortical sources summate at the scalp, so another repetition sensitive ERP, generated elsewhere in the cortex, may be responsible for the observed priming effect at posterior electrodes. We thus used source dipole analysis to test whether the SPN priming effect happens *within* the extrastriate symmetry network, as assumed. A source dipole model was constructed using a sequential strategy [22,23] whereby equivalent current dipoles (ECDs) were fitted to explain the 3D source currents contributing to the observed data [24,25]. Classical LORETA recursively applied (CLARA) [26] was used as an independent source localisation method to determine whether results converge across different localisation methods. The analysis used here was based on other recent work [27].

A source dipole model comprising of two bilateral sources within the extrastriate regions explained 94.7% of variance. Both ECD1 (left Brodmann area 19; approximate Talairach coordinates–x = -26.9, y = -75.9, z = -12.9) and ECD2 (right Brodmann area 19; approximate Talairach coordinates–x = 26.9, y = -75.9, z = -12.9) were located within the fusiform gyrus. The nearest local maximum detected using CLARA was 18.41 mm from both ECD1 and ECD2, evidencing the reliability of the final model. Thus, it appeared that this extrastriate region was the only significant generator of symmetry-specific cortical activity. The final model is detailed in Fig 4A, and the resulting source waveforms for each ECD are illustrated in Fig 4B. In these source waveforms, there was again selective SPN priming effect in the different reflections condition, replicating the sensor level analysis.

Apparent source waveform effects were confirmed with repeated measures ANOVA [Sequence position (first, second, third) X Sequence type (different reflections, identical reflections) X Hemisphere (left (ECD1), right (ECD2)]. The apparent right lateralization in Fig 4C was not significant (F (1,47) = 3.651, p = 0.062). However, there was a main effect of Sequence position (F (1.584, 74.426) = 5.375, $p = 0.011$, $\eta_p^2 = 0.103$), as well as a Sequence position X Sequence type interaction (F (1.712,80.443) = 4.407, $p = 0.020$, $\eta_p^2 = .086$). There was an effect of Sequence position in the different reflections condition (F (1.456, 68.453) = 11.731, p < 0.001, $\eta_p^2 = 0.200$), but not in the identical reflections condition (F (1.731, 81.355) = 1.138, p = 0.319). There were no other effects or interactions (p > 0.211).

In sum, the selective priming effects originally observed in sensor level analysis were found within this pair of bilateral extrastriate sources. From this, we conclude that the SPN priming effect happens *within* the extrastriate symmetry network and does not reflect effects generated elsewhere in the brain.

## Experiment 1 discussion

Experiment 1 found an SPN priming effect for the sequence of three different reflections. Conversely, there was no SPN priming effect for the sequence of three identical reflections. The results were similar for complex and simple patterns, despite their disparate image statistics.

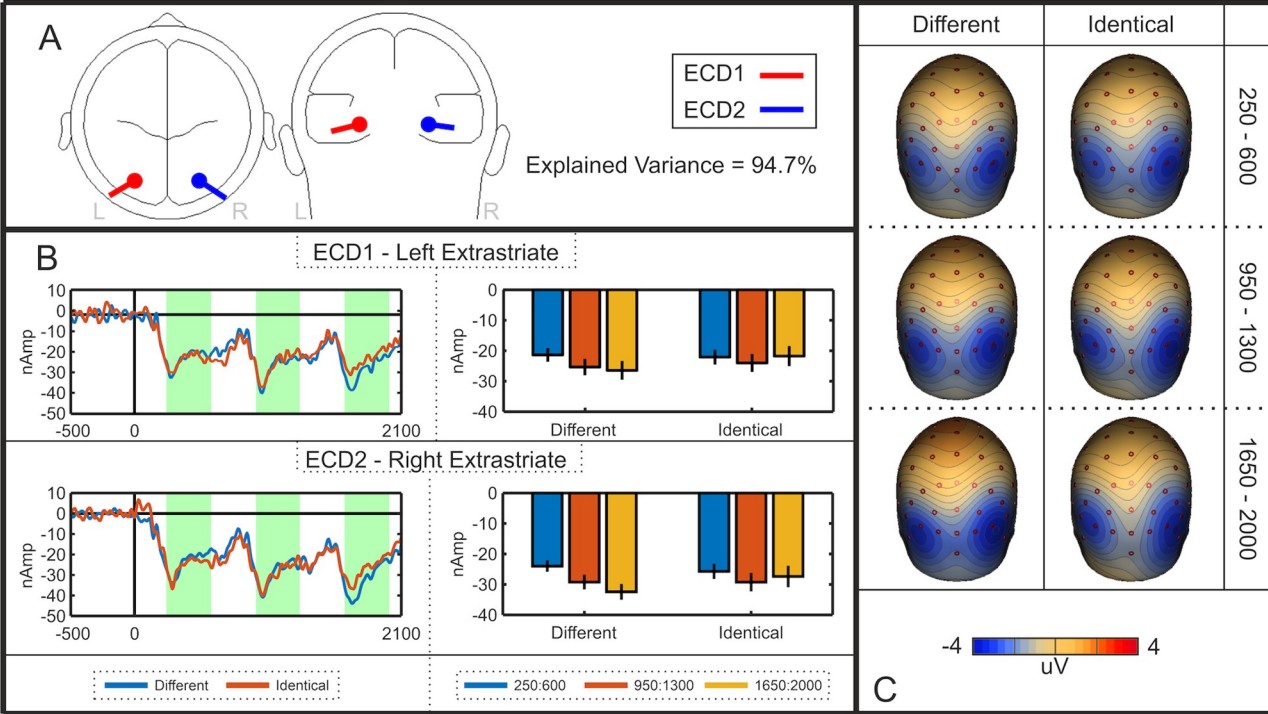

**Fig 4. Experiment 1 source dipole model. A)** Approximate locations of Equivalent Current Dipoles (ECDs) in the left and right extrastriate cortex. **B)** Left and right ECD source waveforms for the different and identical reflection sequences. The intervals used for statistical analysis are shown in green. Bar graphs indicate the mean activity in these intervals (error bars = +/- 1 SD). Note the selective SPN priming effect in these extrastriate source waveforms. **C)** Rear view scalp difference maps (Different Reflection–Different Random; Identical Reflection–Identical Random). Each row shows topographies from first, second and third intervals.

We thus conclude that both SPN priming and the SPN itself are unaffected by large variation of spatial frequency and other low-level visual features.

We also conclude that SPN priming happens *within* the extrastriate network and represents increasing amplitude of the extrastriate symmetry response. Two alternative explanations are less likely. First, the SPN priming effect could also be due to scalp summation from a third repetition-sensitive dipole, which beings later in the trial. However, source-dipole analysis did not support this alternative. Second, the different exemplar sequences might evoke cognitive surprise. Indeed, several ERPs, such as the Mismatch Negativity, are generated when expectations build up over a sequence of events and are then violated [28]. However, such non-specific expectation effects are subtracted when the SPN is computed as a difference wave (different reflections–different randoms). It is also noteworthy that different random and identical random waves were very similar (Fig 3A), which is inconsistent with cognitive surprise explanations.

It seems that there is something special about the onset of new reflection exemplars, which increases activation of the extrastriate SPN generators. This is consistent with some behavioural findings from Sharman and Gheorghiu [14], who found that symmetry discrimination improved when reflected elements moved to new positions or rapidly flickered on and off. Niimi et al. [15] also found a perceptual advantage for dynamic stimuli.

Another recent SPN study by Bertamini, Rampone, Oulton, Tatlidil, & Makin [29] found comparable results. In this experiment two patterns were separated by a one second ISI. The SPN generated by the second pattern was larger in the category repeat condition (different

examples of reflectional symmetry) than in the exemplar repeat condition (identical examples of reflectional symmetry). This again shows that SPN priming only happens when novel exemplars are repeated. The fact that this experiment involved different task and stimuli provides good converging evidence for selective SPN priming.

Having established the basic SPN priming effect with changing exemplars, subsequent experiments exploited the effect to assess overlap and independence of the visual regularity code. If SPN priming transfers between conditions, then conditions may share processing resources. Experiment 2 tested whether SPN priming transfers across changes in retinal location.

## Experiment 2: Repeated and changing retinal locations

Experiment 2 tested whether SPN priming transfers between left and right visual hemifields. We expected no inter-hemispheric transfer of SPN priming. Indeed, there is some experimental evidence to suggest that regularities presented in different retinal locations are functionally independent. For instance, Wright, Makin and Bertamini [30] presented symmetrical patterns on the left or right of fixation, and recorded an SPN in the contralateral hemisphere only. This contralateral SPN was not altered by the pattern presented to the ipsilateral hemisphere (symmetry, asymmetry or nothing).

Experiment 2 used the same 8-fold reflections as Experiment 1. On each presentation in the triplet, a pair of patterns was presented with one pattern on either side of central fixation (therefore, a trial involved a triplet of pairs). Patterns were 5˚ in diameter, and the gap between the left and right patterns was also 5˚ (Fig 5). This ensured that symmetry information was presented in the opposite hemifield, well outside the putative foveal confluence. Participants fixated centrally, and fixation was monitored online with an eye tracker. We analysed eye data to confirm that participants did not routinely break central fixation or move their eyes towards the symmetrical patterns (see Supplementary materials 2).

SPNs were computed as difference from the double random condition (black framed triplets in Fig 5). The crucial comparison was between *repeated* location conditions, where all three reflections were on one side (either the left or right, red framed triplets in Fig 5), and *changing* location conditions, where reflection switched from left to right or vice versa (green framed triplets in Fig 5). Note that the total amount of reflection presented in a trial was identical in repeated and changing location conditions, therefore, if the extrastriate network codes regularity independently of retinal location, then SPN priming effect would be the same in both.

### Experiment 2 methods

Another 48 participants were involved in Experiment 2 (age 16–43, mean age 22, 11 male, 8 left-handed). We tested 24 participants in the simple pattern condition and 24 in the complex pattern conditions. There were 30 blocks of 14 trials (120 random trials, and 60 of each of the other 4 conditions). There were 60 additional oddball trials (14.285%) which were not included in EEG analysis. Participants performed the same oddball discrimination task as in Experiment 1, although blank oddballs now involved presentation of two blank disks. Participants gave the correct answer in 96% of trials. The same time windows and electrodes were used in ERP analysis as Experiment 1.

### Experiment 2 results

Results from repeated and changing location conditions are shown in Fig 6. The difference waves in Fig 6B and 6C represent the difference from the double random condition (where

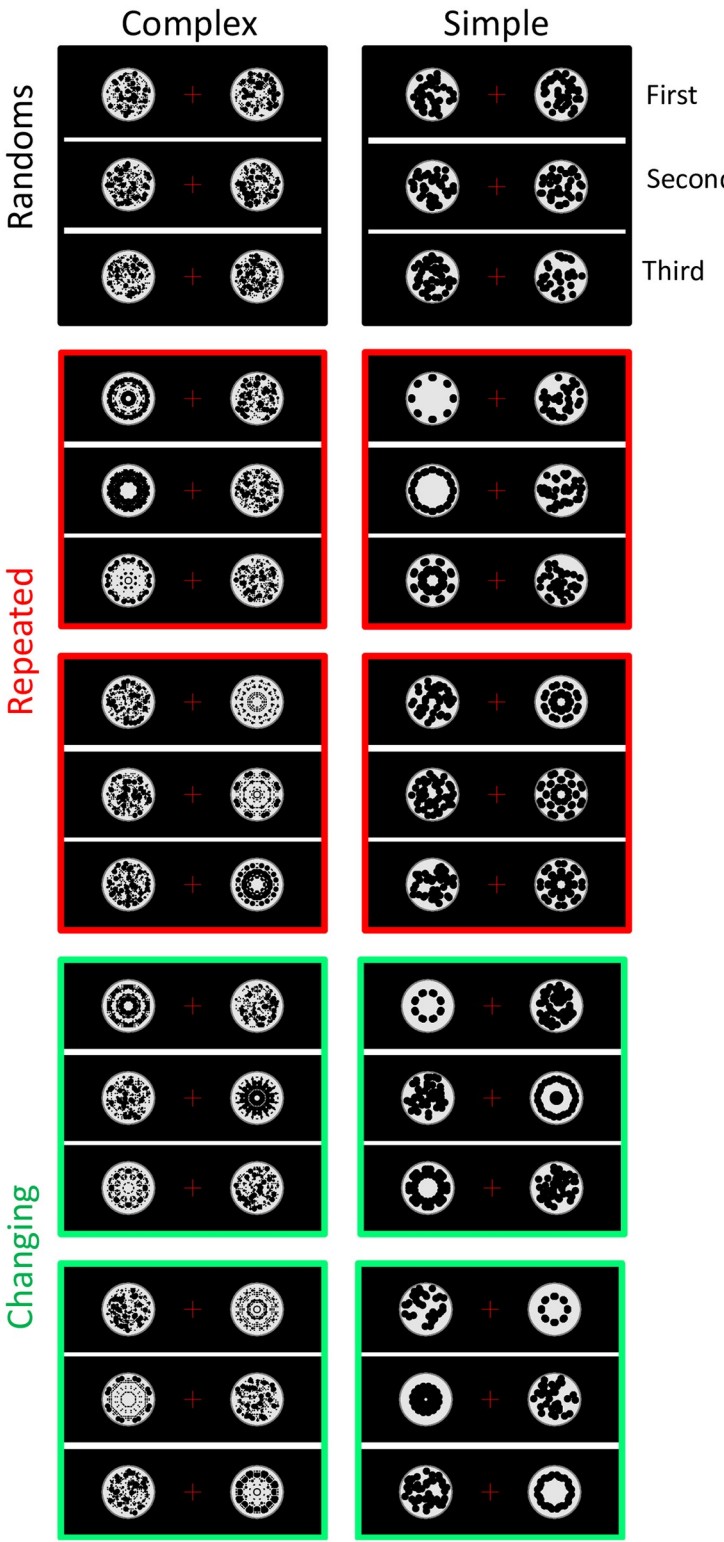

**Fig 5. Experiment 2 stimuli.** Each triplet had three pairs, with one pattern on the left of fixation, and one on the right of fixation. 24 participants were presented with complex pattern pairs (left column) and another 24 were presented with simple pattern pairs (right column).

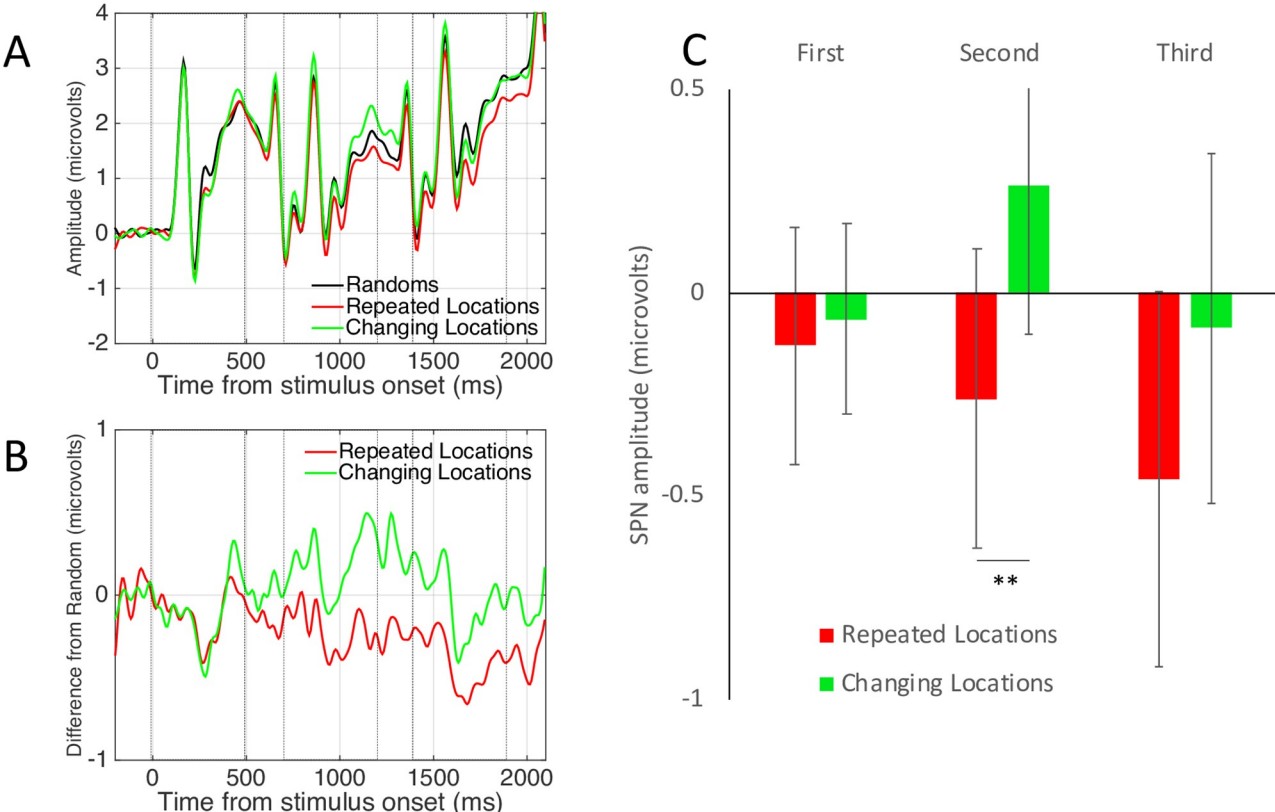

**Fig 6. Experiment 2 results.** Conventions are the same as Fig 3. A) Grand Average ERPs from electrode cluster O1, PO7, O2 and PO8. B) Difference waves as difference from the double random condition. C) Mean SPN for first, second and third intervals. Error bars = 95% confidence intervals. Stars show significant pairwise comparisons (p <0.01) Note SPN priming in the repeated locations condition (red), but not in the changing locations condition (green).

both left and right patterns were random). There was a clear SPN priming effect when the location of reflections was repeated, but not when location of reflections alternated between visual hemifields. None of the 6 SPNs in Fig 6C was significant (one sample t-test, $p > 0.05$). This SPN is not surprising. Our analysis was optimised to assess interhemispheric transfer of SPN priming: It thus averages over left and right hemisphere electrodes (a more detailed breakdown of results by hemisphere is available in Supplementary materials 3).

A mixed ANOVA confirmed these impressions. There were two within-subject factors [3 Sequence position (first, second, third), Sequence Type (repeated locations, changing locations)] and one between-subjects factor [Pattern complexity (complex, simple)]. This found main effects of Sequence position (F (2,92) = 3.296, p = 0.041, $\eta_p^2 = 0.067$) and Sequence Type (F (1,46) = 5.377, = 0.025, $\eta_p^2 = 0.105$) and a Sequence position X Sequence type interaction (F (1.698, 78.096) = 5.410, $p = 0.009$, $\eta_p^2 = 0.105$). As expected, there was a significant linear effect of Sequence position in the repeated locations condition (F (1,47) = 4.575, $p = 0.038$, $\eta_p^2 = 0.089$), but not in the changing locations condition (F < 1). There were no other effects or interactions (F < 1).

Unfortunately signal-to-noise ratio in Experiment 2 was not sufficient for source dipole analysis. Complementary mass univariate and GFP analyses are included in Supplementary Materials 1.

### Experiment 2 discussion

Experiment 2 replicated the contralateral SPN from Wright et al. [30]. As predicted, there was an SPN priming effect when reflectional symmetry was repeated in the same hemifield, but not when it switched between left and right visual hemifields. This suggests that there are independent symmetry sensitive networks in each cerebral hemisphere, and that cross talk between the hemispheres is minimal.

Nevertheless, these results are complicated slightly by an ipsilateral contribution to SPN priming in the repeated locations condition (as described in Supplementary materials 3). Specifically, when reflection was repeated the left hemifield, there was SPN priming in the contralateral right hemisphere (as expected), but also increasing negativity in the ipsilateral left hemisphere (not expected). This could be explained if participants broke central fixation and moved their eyes to the symmetry on the left side of the screen. However, eye tracking provided no evidence for this (Supplementary materials 2). It would be a mistake to overinterpret this unexpected aspect of the results.

### Experiment 3. Repeated and changing orientations

Wenderoth [31] found that symmetry discrimination can benefit from repetition of axis orientations within a block of trials. Experiment 3 thus examined independence of reflections with different axis orientations. This experiment required one-fold reflection (Fig 7). On the first presentation of the triplet, axis orientation angle was set randomly between 0 (vertical) and 90 (horizontal). In the repeated orientation sequences, the chosen orientation was repeated on the second and third presentations. In the changing orientation sequence, orientation was set randomly again on the second and third presentation (with the constraint that subsequent

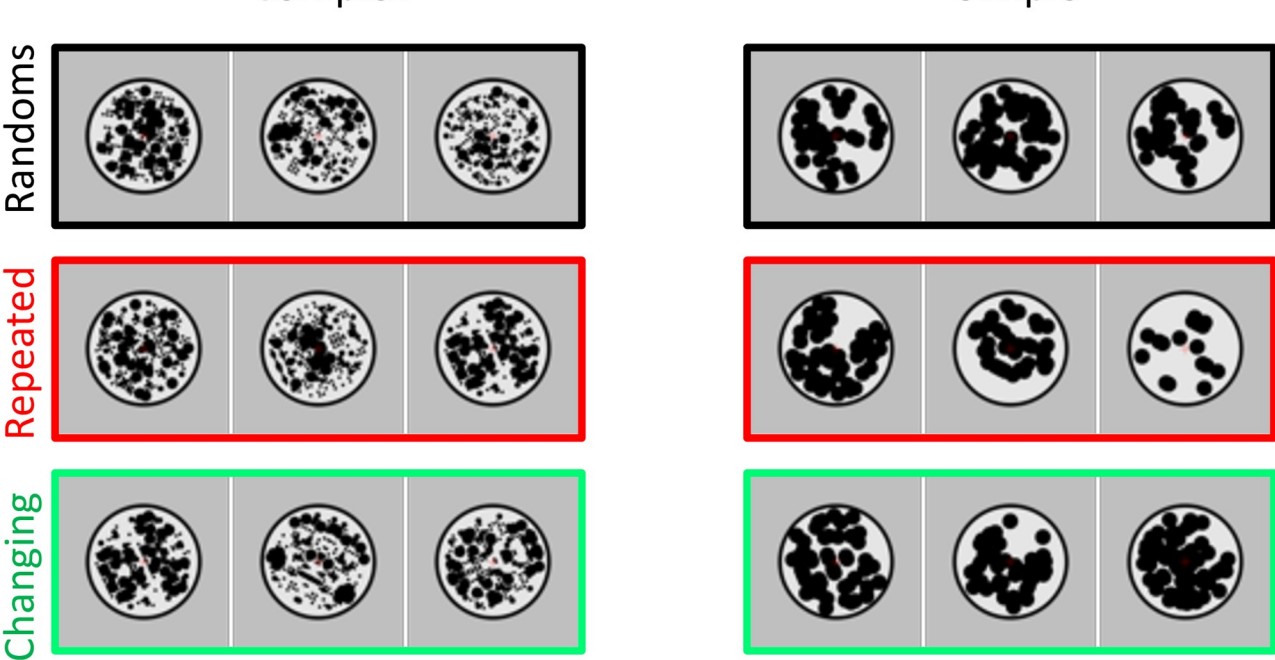

**Fig 7. Experiment 3 stimuli.** 24 participants were presented with complex patterns (left column) and another 24 were presented with simple patterns (right column).

orientations had to be separated by more than 10 degrees). The task was the same as Experiment 1 and 2. All stimuli were presented centrally at fixation.

### Experiment 3 methods

Experiment 3 involved another group of 48 participants (age 18–51, mean age 22, 9 male, 5 left-handed). There were 24 participants in the simple pattern condition and 24 in the complex pattern condition. There were 18 trials in 15 blocks (120 random trials, 60 repeated and 60 changing orientation trials). There were 30 additional oddball trials (11.1%). On average, participants gave the correct response on 96% of the trials. The same time windows and electrodes were used as Experiments 1 and 2.

On consistent orientation trials, the orientation of the first pattern was set at random between 0 (vertical) and 90 degrees (horizontal), then the second and third patterns had the same orientation as the first. On changing orientation and random trials, the orientation of the second and third patterns was again set randomly between 0 and 90 degrees. However, there was a constraint that the absolute angular offset between one pattern and the next had to exceed 10 degrees. This prevented subsequent patterns in the changing sequence from having similar orientations by chance.

### Experiment 3 results

Grand average ERPs from Experiment 3 are shown in Fig 8. All six reflections generated a significant SPN (one sample t tests, $p < 0.005$). A mixed ANOVA revealed no significant main effects ($p > 0.060$) and the predicted Sequence position X Sequence type interaction was not significant (F (1.567, 72.084) = 2.710, p = 0.086). However, there was an expected linear effect of Sequence position in the repeated orientation condition (F (1,47) = 5.838, $p = 0.020$, $\eta_p^2 = 0.110$) but not in the changing orientations condition (F < 1). This suggests that SPN priming does not transfer between orientations that vary unpredictably. Again, the signal to noise ratio was not sufficient for source dipole analysis in Experiment 3. Complementary mass univariate and GFP analyses are included in Supplementary Materials 1.

### Experiment 3 discussion

SPN priming was present in the repeated orientations condition, by not in the changing orientations condition. However, we are cautious about interpreting this apparent selectivity, because there was no significant interaction (for discussion of absent interactions in cognitive neuroscience, see [32]). On the other hand, the results are broadly consistent with those of Wenderoth [31], who found that axis orientation priming enhanced symmetry detection in 2D dot patterns, and with those of Yamauchi et al. [33], who found related results for 3D objects. We can thus be moderately confident that the reflection code is not completely orientation-invariant.

Despite the apparent orientation-selectivity of SPN priming in Experiment 3, we cannot claim SPN priming is completely orientation selective without examining transfer between horizontal and vertical reflections. After all, horizontal and vertical axes have a special visual status [34] and horizontal and vertical reflections are more salient in many conditions (Wenderoth, 1994) [31]. This was the topic of Experiment 4.

## Experiment 4: Vertical and horizontal orientations

Experiment 4 was closely related to Experiment 3, but all reflections were either horizontal or vertical. The repeated orientations condition either involved all horizontals or all verticals. The

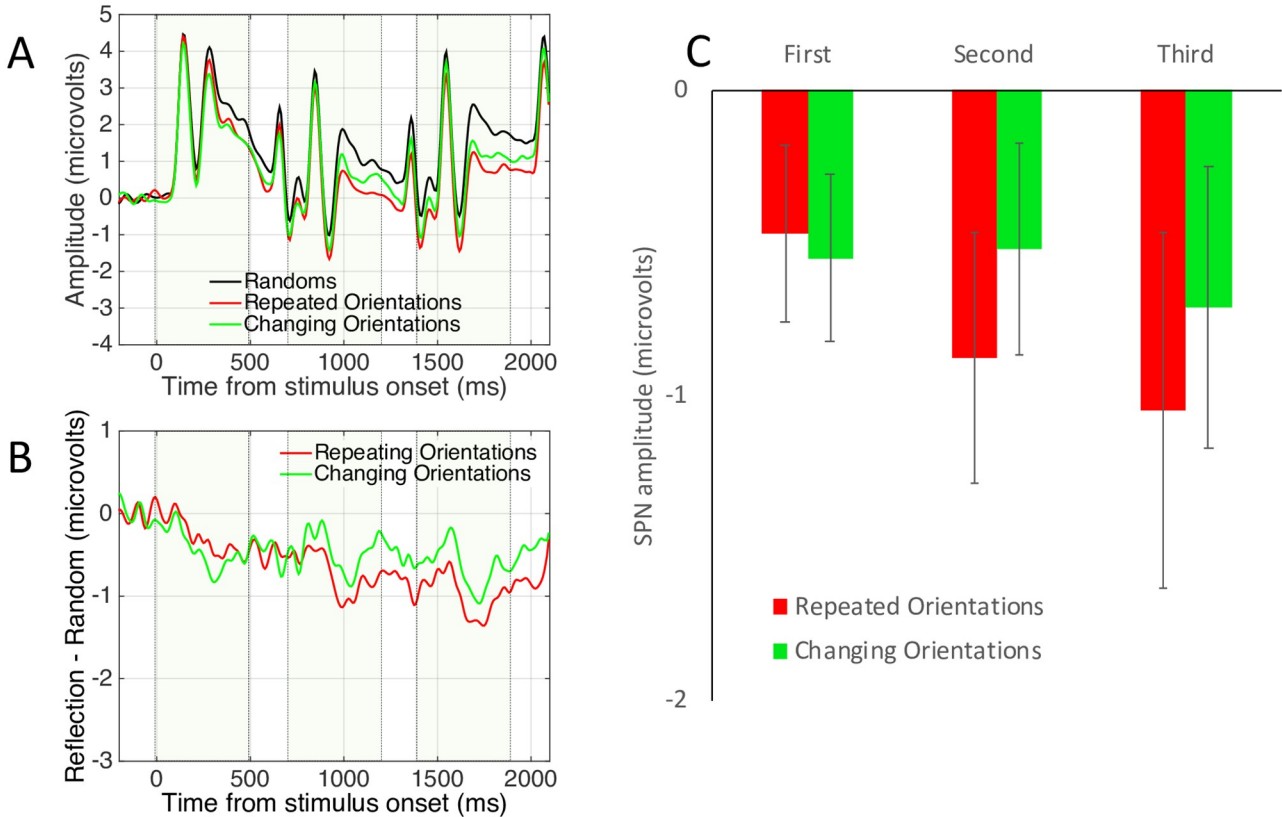

**Fig 8. Experiment 3 results.** Note SPN priming in the repeated orientations condition (Red) but not in the changing orientations condition (green).

changing orientations condition involved switches between horizontal and vertical or vice versa (Fig 9).

Experiment 4 was an important extension of Experiment 3. Previous behavioural research shows that orthogonal axes might be more perceptually coupled than non-orthogonal axes. Treder, van der Vloed and van der Helm [35] examined symmetry discrimination when patterns were preceded by same axis primes, orthogonal axis primes or non-orthogonal axis primes. They found a similar facilitation for same and orthogonal axis primes, but inhibition for non-orthogonal primes. Treder et al.'s result suggests that SPN priming should transfer between orthogonal horizontal and vertical reflections in Experiment 4.

## Experiment 4 methods

Another 48 participants were recruited (age 18–62, mean age 23, 8 male, 2 left-handed). Experiment 4 had the same design as Experiment 3. There were 24 participants in the simple pattern conditions and 24 in the complex pattern conditions. The consistent orientation conditions involved sequences of either three verticals (30 repeats) or three horizontals (30 repeats) (these trials were averaged in EEG analysis). The changing orientation conditions involved sequences of Vertical Horizontal Vertical (30 repeats), or Horizontal, Vertical Horizontal (30 repeats) (these were also averaged in EEG analysis). There were 30 additional oddball trials. On average, participants gave the correct answer in 98% of trials. The same time windows and electrodes were used in ERP analysis as Experiments 1–3.

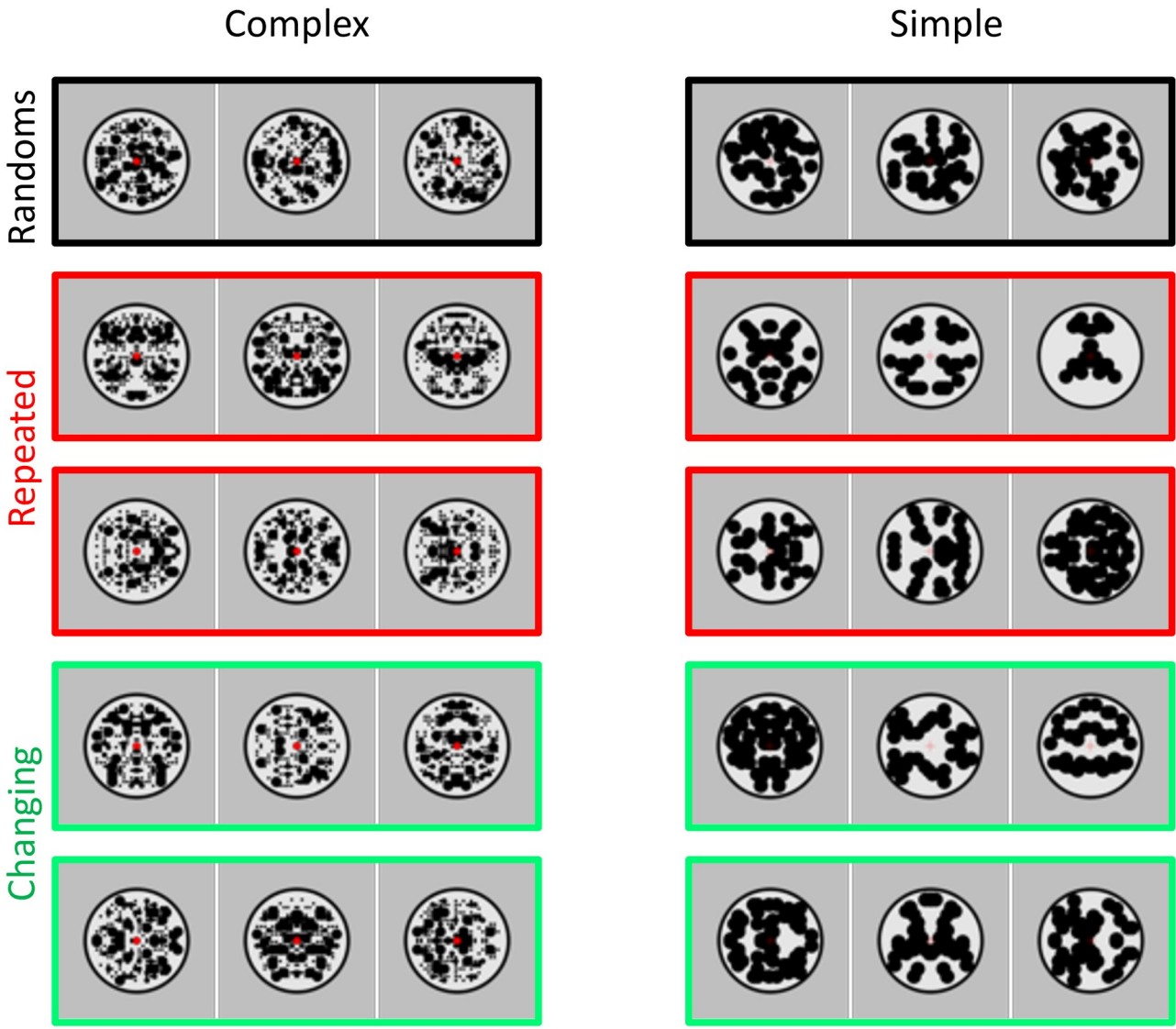

**Fig 9. Experiment 4 stimuli.** 24 participants were presented with complex patterns (left column) and another 24 were presented with simple patterns (right column).

### Experiment 4 results

Results are shown in Fig 10. All 6 conditions generated a significant SPN ($p <0.001$). Unlike Experiment 3, there was an SPN priming effect in both repeated and changing orientations conditions. A mixed ANOVA revealed a significant main effect of Sequence position (F $(1.706, 78.463) = 8.981$, $p = 0.001$, $\eta_p^2 = 0.163$), linear contrast (F $(1, 46) = 11.396$, $p = 0.002$, $\eta_p^2 = 0.199$) but no other effects or interactions ($p > 0.118$). Again, complementary mass univariate and GFP analyses are included in Supplementary Materials 1.

A source dipole model comprising of two bilateral sources within the extrastriate regions explained 76.7% of variance. Both left ECD1 (Brodmann area 19; −x = -24.3, y = -64.7, z = -8.1) and right ECD2 (Brodmann area 19;−x = 24.3, y = -64.7, z = -8.1) were located within the fusiform gyrus. The nearest local maximum to ECD1 and ECD2 detected using CLARA was 15.17 mm and 22.617 mm respectively. The final model is detailed in Fig 11A, and the resulting

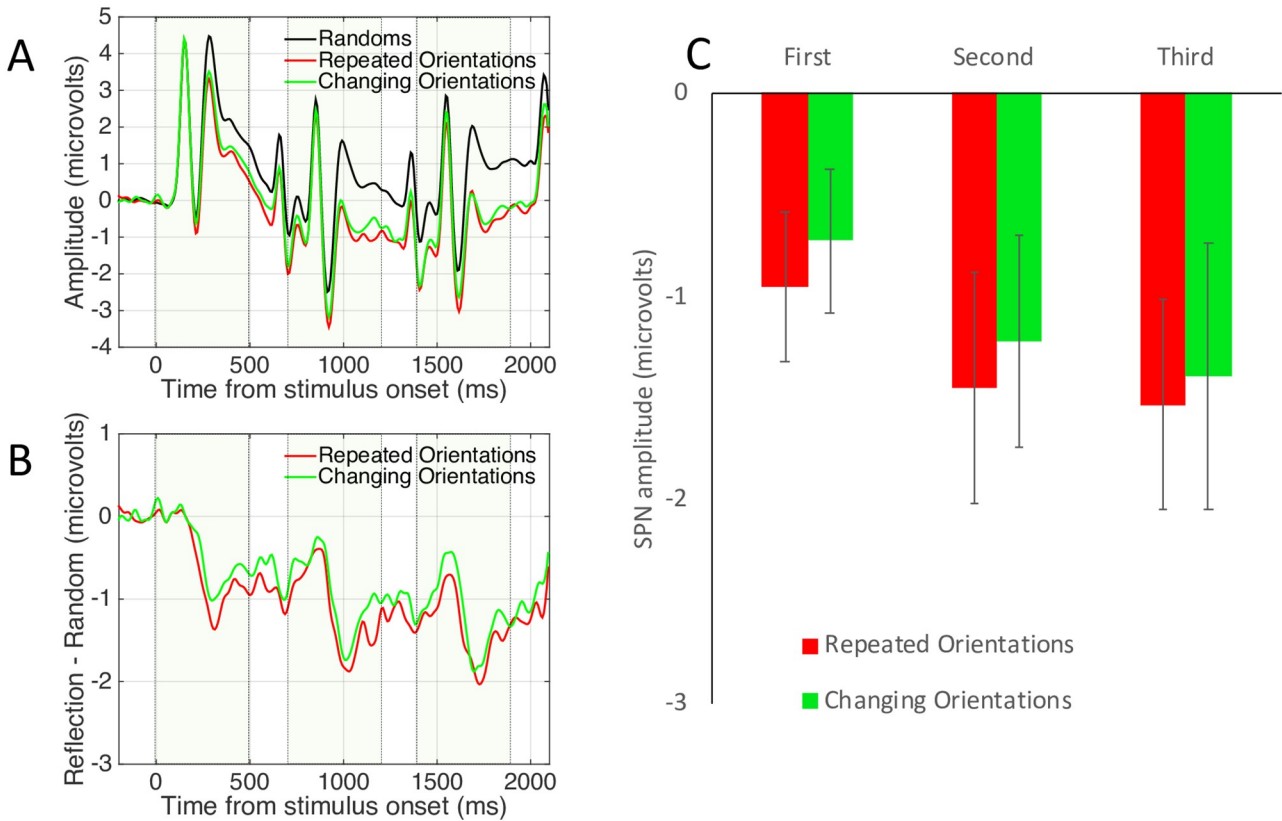

**Fig 10. Experiment 4 results.** Note SPN priming in both repeated and changing orientation conditions.

source waveforms are illustrated in Fig 11B. It can be seen that the SPN priming effect was found in both hemispheres, an in both repeated and changing orientation conditions. This was confirmed by three-way repeated measures ANOVA [Sequence position (first, second, third) X Sequence type (repeated orientations, changing orientations) X Hemisphere (left (ECD1); right (ECD2)], which revealed a main effect of Sequence position (F (1.524,71.603) = 11.362, $p < 0.001$, $\eta_p^2 = 0.195$), but no other effects or interactions (p > 0.229).

### Experiment 4 discussion

Experiment 4 provided the first clear example of transfer: Namely, SPN priming transferred between vertical and horizontal orientations. While we cannot be sure that transfer would be replicated with other orthogonal axis pairs (e.g. 45 and 135 degrees), Experiment 4 suggests that vertical and horizontal axes are perceptually coupled [35].

### Experiment 5: Consistent and changing regularities

Finally, Experiment 5 examined independence of reflection and rotational symmetry. We used 90-degree rotation and 1-fold reflection because they were approximately matched for salience. The consistent regularity condition involved three reflections or three rotations. The changing regularity condition involved sequences where reflection and rotation alternated (Fig 12). We expected reflection and rotation to be independent in Experiment 5.

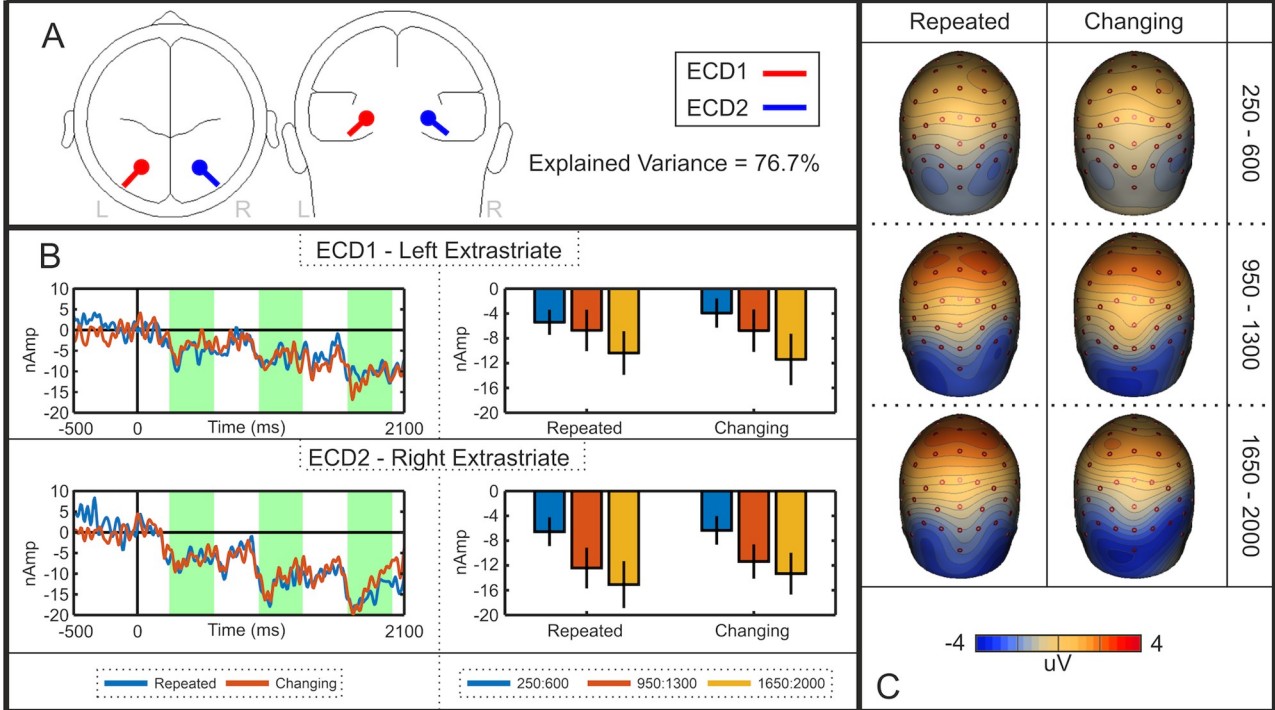

**Fig 11. Experiment 4 source dipole model.** Conventions are the same as Fig 4, albeit with different Y axis scales. Note the ubiquitous SPN priming effect in the left and right extrastriate source waveforms.

## Experiment 5 methods

Another 48 participants were recruited (age 16–52, mean age 25, 12 males, 6 left-handed). As usual, there were 24 participants in the simple pattern condition and 24 in the complex pattern conditions (Fig 12). There were 30 blocks of 18 trials. There were 240 random patterns. There were 60 repeats of each regular sequence type and 60 additional oddball trials (11.1%). On average participants gave the correct answer on 98% of trials. The same time windows and electrodes were used in ERP analysis as Experiments 1–3.

## Experiment 5 results

Results are shown in Fig 13. All 6 conditions generated a significant SPN (one sample t tests, $p <0.001$). There was an SPN priming effect in both repeated and changing regularity conditions. A mixed ANOVA revealed a significant main effect of Sequence position only (F (1.460, 67.169) = 19.388 $p < 0.001$, $\eta_p^2 = 0.298$, linear contrast F (1, 46) = 23.088, $p < 0.001$, $\eta_p^2 = 0.334$). There were no other effects or interactions ($p > 0.128$).

Unlike Experiments 1–4, mass univariate and GFP analyses were not perfectly consistent with the traditional ERP analysis. Specifically, the SPN priming effect in changing regularity condition was not supported by mass univariate analysis (https://osf.io/2yjus/).

As with Experiment 4, a source dipole model comprising of two bilateral sources within the extrastriate regions explained 92.01% of variance. Both left ECD1 (Brodmann area 19;–x = -25.6, y = -60.8, z = -10.2) and right ECD2 (Brodmann area 19;–x = 25.6, y = -60.8, z = -10.2) were located within the fusiform gyrus. The nearest local maximum to ECD1 detected using CLARA was 16.322 mm for ECD1. ECD2 was localized to the same point using CLARA. The final model is detailed in Fig 14. The SPN priming effect was evident in both extrastriate

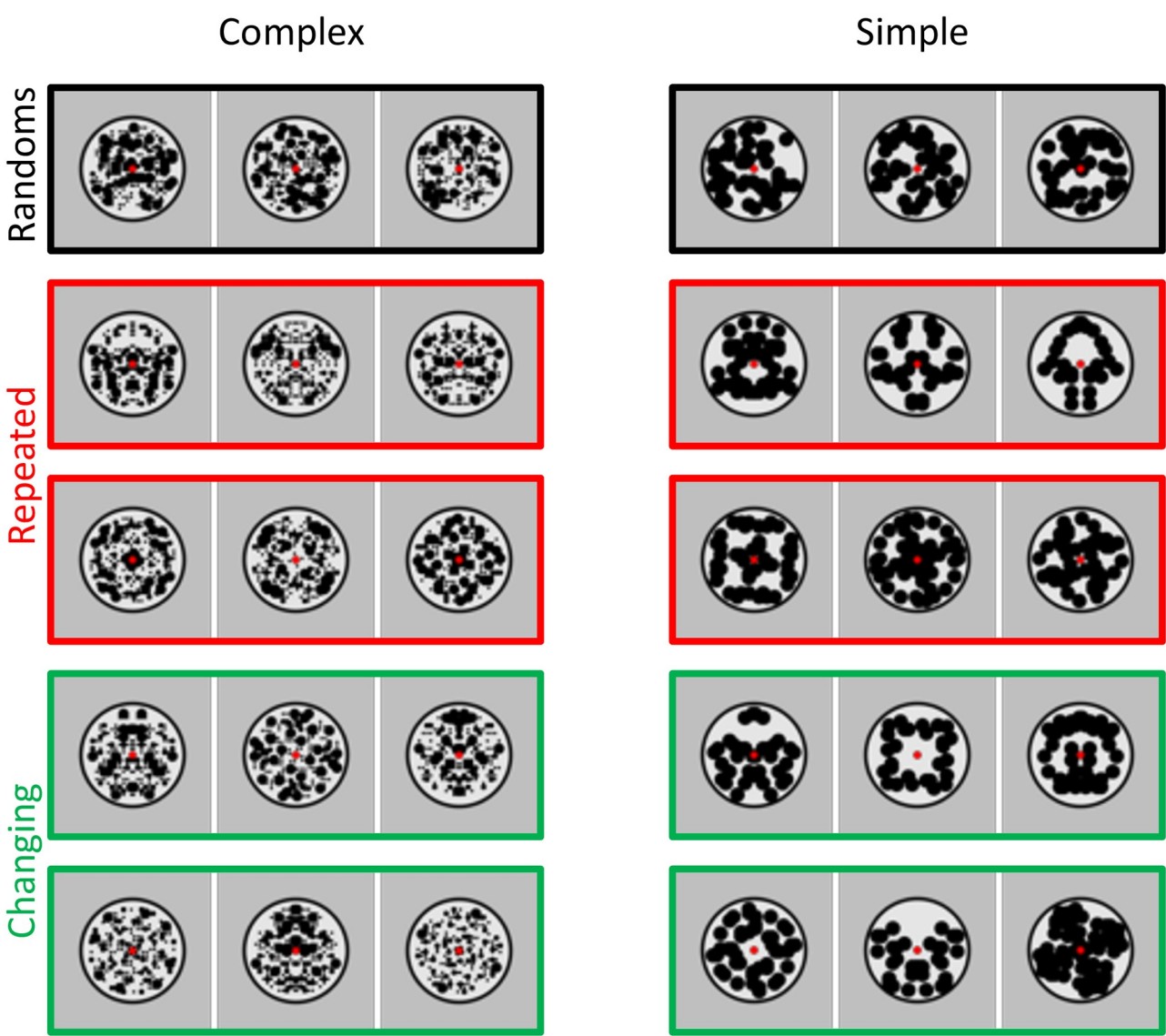

**Fig 12. Experiment 5 stimuli.** 24 participants were presented with complex patterns (left column) and another 24 were presented with simple patterns (right column).

dipoles. This replicates the source waveform analysis in Experiments 1 and 4. However, there was one unexpected effect: the SPN was selectively attenuated in the left hemisphere during the changing regularity condition.

We performed a three-way repeated measures ANOVA [Sequence position (first, second, third) X Sequence type (repeated regularity; changing regularity) X Hemisphere (left (ECD1); right (ECD2)]. There was a main effect of Sequence position (F $(1.633, 76.738)$ = 17.89, $p < 0.001$, $\eta_p^2 = 0.276$). There was an unexpected interaction between Sequence type and Hemisphere (F$(1,47)$ = 4.619, $p = 0.037$, $\eta_p^2 = 0.089$) because the SPN was right lateralized in the changing regularity condition (F $(1,47)$ = 5.811, p = 0.020, $\eta_p^2 = 0.110$) but not in in the repeated regularity condition (F $< 1$). There were no other effects or interactions (p $> 0.204$).

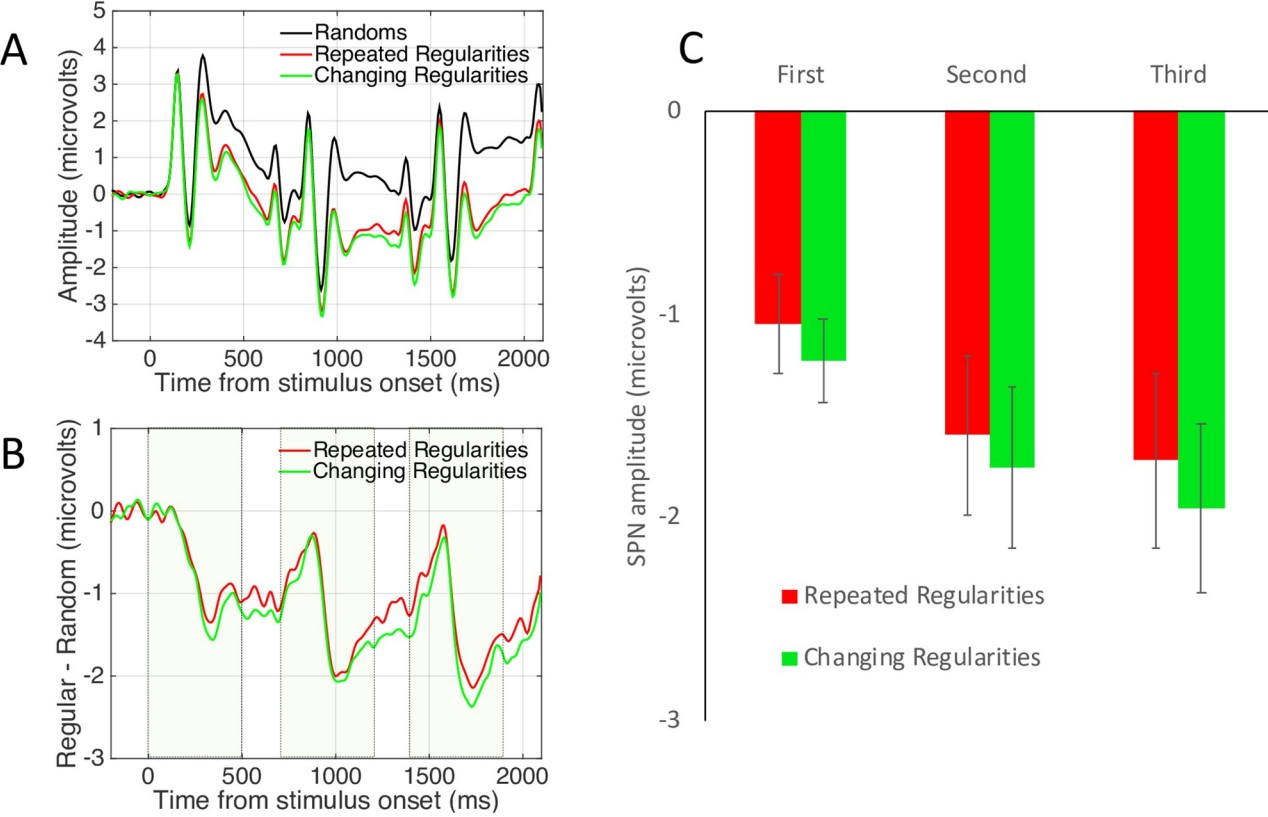

**Fig 13. Experiment 5 results.** Note SPN priming in both repeated and changing regularity conditions.

### Experiment 5 discussion

ERP results of Experiment 5 were surprising. There was an SPN priming effect for repeated regularities, *and* also when regularity alternated between reflection and rotation. This suggests some overlap between the way reflection and rotation are coded in the visual system. Although the traditional ERP analysis was straightforward, source dipole analysis found a relatively weak response to changing regularity in the left hemisphere, and SPN priming for changing regularity was not apparent with mass univariate analysis. Nevertheless, the balance of evidence suggests that SPN priming at least partially transfers from reflection to rotation and vice versa, despite our initial predictions.

### General discussion

In Experiment 1, we presented triplets of patterns and found that SPN amplitude *increased* from presentation $1 > 2 > 3$ when different exemplars were used. This was a case of repetition enhancement within the extrastriate symmetry network. This SPN priming was consistent with previous behavioural studies [14].

Our subsequent experiments exploited SPN priming to test the independence of visual regularity representations. Experiment 2 provided evidence of SPN priming within one hemisphere, and no hint of transfer between hemispheres. Experiment 3 found SPN priming when orientation was consistent, but not when it changed unpredictably. These two experiments suggest different regularity codes are functionally independent. However, Experiment 4 found that SPN priming survived alternation between vertical and horizontal axis orientations.

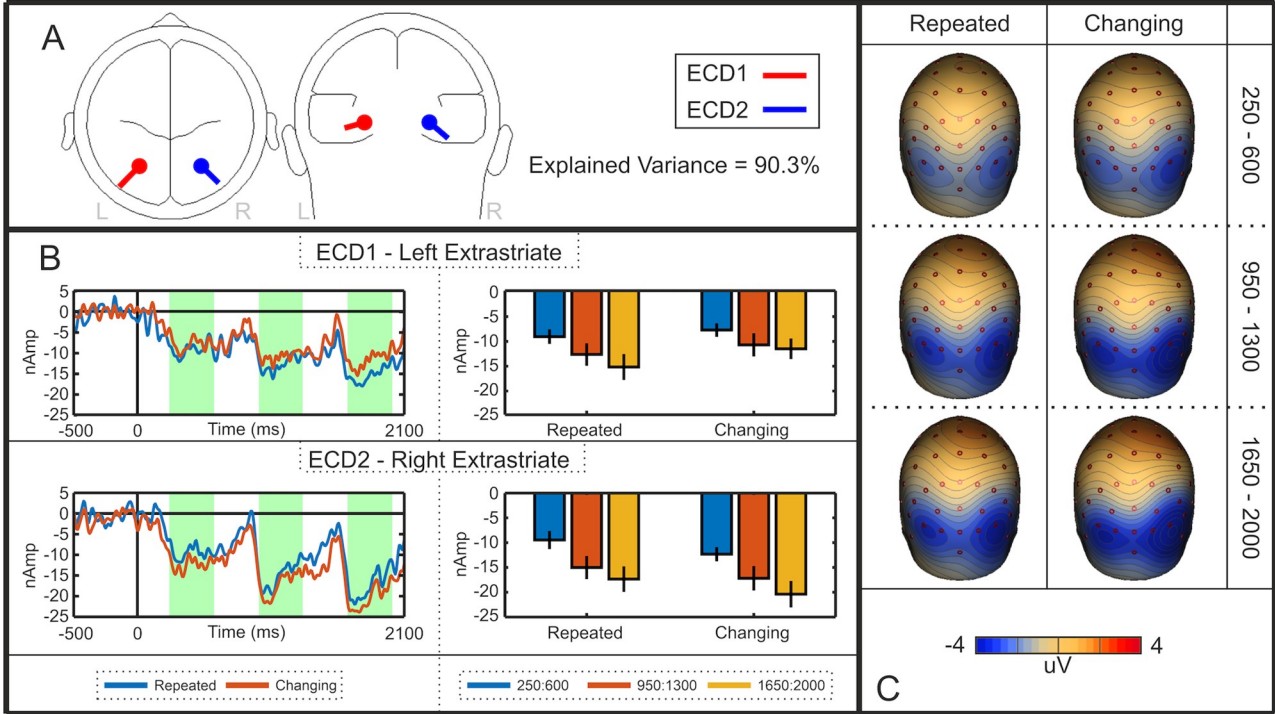

**Fig 14. Experiment 5 source dipole model.** Note the ubiquitous SPN priming effect in these source waveforms, but right lateralization in the changing regularity condition.

Finally, and most surprisingly, Experiment 5 found that SPN priming transferred between reflection and rotational symmetry. These two experiments suggest that different regularity codes partially overlap. In other recently published work, we found SPN priming transferred between black and white reflections. This shows the regularity code is independent of luminance polarity [36] as anticipated by previous work using other methods [37–39]. It seems that different regularities are coded by overlapping neural populations, but only when they are presented at the same retinal location.

Some previous work inspired by filter models of symmetry perception [40] has also addressed this topic of representational independence. It has been found that symmetry discrimination is not impaired by noise masks with different spatial frequencies [41] or orientations [42]. This suggests that the symmetry representations are built on retinotopic maps of low-level features, and therefore different regularity representations with different low-level features do not overlap and perceptually interfere with each other. In contrast, SPN priming suggests that disparate regularity codes *can* overlap—we found some crosstalk between neural systems that code horizontal and vertical reflection (Experiment 4), and between reflection and rotation (Experiment 5). These experiments suggest more representational overlap than filter models allow. Perhaps there are higher-level visual integration mechanisms with properties not captured by the filter models, and SPN priming occurs at this higher level?

## Interpretational considerations

Repetition paradigms are widely used, but there are many important interpretation considerations which can render the results of a single experiment ambiguous [21]. Furthermore, the SPN priming effect is relatively new, so our interpretation necessarily involves some debatable

assumptions. However the results of Experiment 1 are consistent with Bertamini et al. [29] even though that study different procedure and trial structure. Likewise, the results of Experiment 1 are arguably a conceptual replication of Sharman and Gheorghiu [14] and Niimi et al. [15]. This consistency increases confidence in our interpretation of Experiment 1 (and the subsequent experiments which build on this foundation).

We acknowledge that source localization is insufficient to support an original claim about neuroanatomy, especially when it based on just 64 electrodes. However, converging evidence from previous fMRI, TMS and EEG studies make us confident that the SPN is generated in the extrastriate cortex. We do not need the new source localization analysis to be confident that the SPN is generated in the extrastriate cortex. However, we do need source dipole analysis to establish that SPN priming recorded on the scalp actually happens within the extrastriate cortex. Without the source dipole analysis, our interpretation of SPN priming would be ambiguous.

We note that the SPN priming effect, when present, was always around 0.5 μV, whatever the amplitude of the SPNs themselves (e.g. an increase from -2.6 to -3.3 μV in Experiment 1, but only -0.47 to -1.05 μV in Experiment 3). This might imply that SPN priming is caused by some non-specific increase in visual alertness. However, there was no SPN priming in Experiment 1 when identical patterns were repeated, so this explanation is less likely.

Most EEG and fMRI repetition studies have used pairs of stimuli, rather than triplets. With hindsight, this two-repeat convention is probably superior, because the third presentation is inherently complicated (It is a repeat of the first presentation even in changing sequences, and it is cognitively predictable in repeated sequences). However, the potential disadvantages of triplets were not a large issue in practice: Most of the repetition effects found here were already evident in the first two repeats, and none were reversed on the third repeat.

Our participants were not attending to regularity and explicitly classifying the patterns as symmetrical or random. It could be that the SPN priming becomes more transferrable when participants attend to regularity. Alternatively, the distinctions between different types of regularity could be sharpened when participants attend them, so SPN priming could become *less* transferable. This ambiguity can only be resolved with future experiments.

Next, we note that there are some inconsistencies between repetition effects measured with different techniques. For instance, stimulus repetition effects on single cell responses in IT seem to be more selective that those measured in classic fMRI adaptation (fMRI-A) paradigms [43]. Likewise, face repetition effects have not always been consistent from fMRI-A to ERP studies [20]. Many SPN results mirror fMRI results [6], so one might expect repetition enhancement in the BOLD symmetry response with repeated presentation. On the other hand, BOLD repetition suppression is caused by repeated presentation Glass patterns [17]. Whether the BOLD response to symmetry is subject to repetition enhancement or suppression is again an empirical question.

Finally, abstract symmetrical patterns might be perceived as faces (a case of face pareidolia). However, it is unlikely that activation of face-sensitive visual areas with repeated presentation is responsible for SPN priming. The face response would be strongest for 1-fold vertical reflection in Experiment 4, and SPN priming was not unique to this condition.

## Future work on SPN priming

As mentioned, Makin et al. [36] used SPN priming technique to demonstrate that the regularity code is luminance independent. Future work could use the SPN priming effect to answer other questions about symmetry coding. For instance, people can perceive *anti-symmetry*, where luminance in symmetrical positions is *anti-correlated* (e.g. black regions paired with

white and white regions paired with black). We know that anti-symmetry generates an SPN, albeit with slightly reduced amplitude [39,44]. However, it is uncertain whether symmetry and anti-symmetry are coded by the same extrastriate networks [38]. Future work could examine this by testing whether SPN priming generalizes from symmetry to anti-symmetry. Finally, future work could measure SPN priming across changes in virtual view angle [45]. Perhaps the symmetry code is view invariant under some conditions?

## Conclusions

The SPN priming effect is robust when different reflection exemplars are presented. This confirms that dynamism advantages found in behavioural work on symmetry perception. SPN priming is sometimes selective, and it does not survive changes in retinal location or non-orthogonal changes in axis orientation. However, SPN priming is not completely selective, and it transfers between horizontal and vertical orientations and between reflection and rotation. Based on available evidence, we conclude that there are common visual integration mechanisms which can extract *different* regularities at *specific* retinal locations.

## General methods

### Participants

Each of our 5 experiments had 48 participants (240 in total). There were 24 participants in the simple pattern condition and 24 in the complex pattern conditions. All participants had normal or corrected to normal vision, except for those Experiment 2 where people with glasses or contact lenses were excluded because it interfered with eye tracking. The study was approved by the Health and Life Sciences Research Ethics Committee (Psychology, Health and Society) at University of Liverpool (Ref 2122) which ensured that the study in accordance with APA ethics codes. Participants gave written informed consent and signed consent forms in the presence of the researcher.

### Apparatus

All EEG data was collected with a BioSemi Active-2 64 channel EEG system. Electrodes were embedded in an elasticated cap and arranged according to the international 10–20 system. Bipolar HEOG and VEOG channels were monitored for excessive blinking or eye movements and sample rate was 512 Hz. HEOG and VEOG were not included in the montage.

In all Experiments a chin rest was used for head stabilization. Experiments 1, 3 and 4 were presented on a 40 X 30 Cm (23 X 17˚) CRT monitor. Participants were seated 100 cm back from the monitor. In Experiment 2 participants were seated 57 cm back from a 53 X 30˚ LCD monitor. In Experiment 5, participants were seated 57 Cm back from a 51 X 29˚ LCD monitor. All monitors had a refresh rate of 60Hz. Although the monitors changed between experiments, stimulus size was identical in degrees of visual angle.

All Experiments except Experiment 2 were conducted in an electrically shielded and darkened cubicle. Experiment 2 was conducted on an identical BioSemi EEG recording system, but in a separate eye tracking cubicle that was not electrically shielded. In Experiment 2, eye position was monitored with a desk mounted Gazepoint 60Hz infra-red eye tracker. The experimenter could monitor eye position superimposed on the stimulus in real time, and thus assess whether participants routinely broke fixation. Eye position analysis confirmed that participants were fixating successfully without bias to the symmetrical pattern (see Supplementary materials 2).

## Stimulus construction algorithm

The images were constructed using Python and PsychoPy [46], and saved as PNG image files. The algorithm for construction of 8-fold reflection or matched random patterns in Experiment 1 is shown in Fig 1. The patterns were based on an implicit grid of 432 cells. This produced 8 segments, each with a central mirror reflection (Fig 15A). The grid was 6.1˚ of visual angle in diameter, and the outer black ring was 7.2˚ in diameter.

Next half of each segment was occupied with dots in a probabilistic fashion (Fig 15B). For complex patterns, each cell had a 50% probability of occupation. When occupied, small (0.18 ˚ diameter), medium (0.36˚ diameter) or large (0.72 ˚ diameter) dots were chosen at random with from a distribution with a ratio of 6:2:1. In other words, on average, there were 3 times as many small dots as medium, and twice as many medium as large (again this was probabilistic, so it would be possible, albeit extremely unlikely, for all dots to be the same size). For regular patterns, the arrangement was mirrored in each segment, and all segments were identical (Fig 15C). For random patterns, each segment was independent, and the dots on each side were not mirrored. The average number of small, medium and large dots was the same in the regular and random patterns. However, the dot-size variability between patterns was higher for regular patterns because only one segment was original, so any statistically usual properties became like frozen accidents in the other segments.

For simple patterns, the construction algorithm and implicit grid were the same, the probability of occupation was set at 10%, and all dots were 1˚diameter. This results in stimuli with a greater distribution of energy at lower spatial frequencies, larger 'blobby' substructures, and reduced coastline of black regions.

Stimulus construction was the same for Experiments 2–5, except the implicit grid was changed. In Experiment 2 the images were resized, so the black ring was 5˚ in diameter, but the proportional sizes of elements were the same as all other experiments. One-fold reflections was based on an implicit grid with a single vertical seam, and 90-degree rotations had 4 seams. In Experiments 3 and 4, the images were rotated after generation. In Experiment 5 were 120 random triplets based on the 1F grid, and 120 based on the 4F grid. This made no discernible difference to the appearance of the random patterns.

The required number of regular and random images were generated in advance, so no pattern was ever presented twice. In each experiment, all participants were shown the same set of images. However, images were shuffled so that they played different roles for each participant. For instance, a particular reflection might appear as the first in a reflection triplet for one participant, as the third pattern in a RandRandRef sequence for another participant. This abolished any potential effects of chance stimulus features on our results.

## Basic ERP analysis

EEG analysis was based on our previous SPN studies. EEG data was analysed offline using eeglab 13.4 functions in Matlab 2014b [47]. Data from the 64 scalp electrodes was re-referenced to the scalp average, and low pass filtered at 25Hz. We then downsampled to 128 Hz to reduced file size and segmented the data in epochs from -0.5 to + 2.1 seconds around the onset of the first pattern. ERP waves were baseline corrected to a 200 ms pre-stimulus interval.

Independent Components Analysis [48] then used to remove blink and other high amplitude artefacts. The number of ICA components removed was as follows (out of a maximum of 64): Experiment 1: mean = 9.75, min = 0, max = 20, Experiment 2 mean = 12.35, min = 4, max = 22, Experiment 3: mean = 11.46, min = 2, max = 21. Experiment 4 mean = 9.92, min = 2, max = 13: Experiment 5: mean = 9.06, min 2, max = 21. After cleaning with ICA, epochs where amplitude exceeded +/- 100 µV at any electrode were excluded. Mean exclusion

**Fig 15. Stimulus construction steps.** A) An in implicit grid of possible locations was created. B) The grid was probabilistically occupied with black dots. For complex stimuli, the black dots were small, medium or large (top two rows). For simple stimuli, all black dots were large (lower two rows). C) For random patterns each segment half was occupied independently. For reflectional pattern, each half was mirrored.

rates were as follows: Experiment 1; 11–14%, Experiment 2: 14–15%, Experiment 3: 14–15%, Experiment 4: 15–16%, Experiment 5: 8–9%.

All ERP analysis was conducted on electrode cluster [PO7, O1, O2 and PO8]. This was consistent other recent SPN papers [7]. The included trials in each condition were averaged to provide the data for ERP analysis with mixed ANOVA. The Greenhouse-Geisser correction

factor was used whenever the assumption of sphericity was violated (p < 0.05). Consequently, adjusted degrees of freedom are reported throughout. Across the 5 Experiments we analysed 33 SPN difference waves (i.e., each bar in the results figures). None of 33 SPN waves violated the assumption of normality according to the Kolmogorov-Smirnov test of normality ($p < 0.05$). This replicates previous experiments, where individual participant SPN amplitudes are usually normally distributed around the grand average. ERP data used for plotting and statistical analysis is also available on OSF (https://osf.io/2yjus/).

## Source dipole modelling

In order to investigate the spatiotemporal dynamics of SPN priming, a source dipole model was constructed in BESA v. 7.0 (MEGIS GmbH, Munich, Germany) for each experiment. For the greatest accuracy of source localization, it is necessary to utilize data with a large signal-to-noise ratio. To achieve this, difference waves (symmetry–random) for each condition were averaged to produce a single grand-average waveform representing symmetry-specific responses. This was done for each experiment individually.

The protocol for producing an appropriate source dipole model required that equivalent current dipoles (ECDs) were fitted to describe the 3-dimensional source currents in the regions contributing predominantly to the data. Principle component analysis (PCA) was first used to identify an appropriate number of ECDs to fit. Since previous studies have identified bilateral extrastriate cortices as being the primary generators of symmetry specific neural activity [9–11] two ECDs were first inserted in the bilateral extrastriate regions. Following the insertion of these two ECDs, residual variance was used as a tool for indicating the sufficiency of the model. The ECD fitting procedure required waveforms with a large signal-to-noise ratio and intervals with a strong cortical response. Therefore, the weak SPN observed in Experiment 2 and 3 could not be accurately modelled. Since data across experiment 1, 4 and 5 were sufficiently explained by bilateral extrastriate ECDs, no further fitting of ECDs were required.

Classical LORETA analysis recursively applied (CLARA), which is an iterative application of the LORETA algorithm [26] was used to confirm and adjust the locations of the ECDs in the final model. Following the fitting of the ECD locations, the orientation of the ECDs then had to be determined. Since there are differences between individuals regarding gyral anatomy in the brain, ECD orientation was determined on a subject-by-subject basis, but with fixed location between subjects, based on the entire corresponding grand average difference waveform. A 4-shell ellipsoid head volume conductor model was employed using the following conductivities (S/m = Siemens per meter): Brain = .33 S/m; Scalp = 0.33 S/m; Bone = 0.0042 S/m, Cerebrospinal Fluid = 1 S/m. Source waveforms for each experiment and condition were exported and analyzed using repeated-measures ANOVAs.

## Acknowledgments

We would like to thank the University of Liverpool students who helped with data collection: Husnara Ali, Jamie Benyon, Megan Bowen, Faye Clancy, Rebecca Davies, Scarlett Griffin, Adie Howarth, Rebecca Rogers, Nathan Rooney. We would also like to thank the Internship students on summer placement from the University of Calgary: Aya Ebdalla and Emily Drake (2018) and Haya Bakour and Jessica Hsieh (2019).

## Author Contributions

**Conceptualization:** Alexis D. J. Makin, Giulia Rampone, Marco Bertamini.

**Data curation:** Alexis D. J. Makin, John Tyson-Carr, Yiovanna Derpsch.

**Formal analysis:** Alexis D. J. Makin, John Tyson-Carr.

**Funding acquisition:** Alexis D. J. Makin.

**Investigation:** Alexis D. J. Makin.

**Methodology:** Alexis D. J. Makin.

**Project administration:** Alexis D. J. Makin.

**Resources:** Alexis D. J. Makin, Marco Bertamini.

**Software:** Alexis D. J. Makin.

**Supervision:** Alexis D. J. Makin.

**Validation:** Alexis D. J. Makin.

**Visualization:** Alexis D. J. Makin.

**Writing – original draft:** Alexis D. J. Makin.

**Writing – review & editing:** Alexis D. J. Makin, John Tyson-Carr, Yiovanna Derpsch, Giulia Rampone, Marco Bertamini.

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
