## [Decision Letter · Decision Letter 0]

8 Apr 2021

PONE-D-20-34598

Electrophysiological priming effects demonstrate independence and overlap of visual regularity representations in the extrastriate cortex.

PLOS ONE

Dear Dr. Makin,

Thank you for submitting your manuscript to PLOS ONE. After careful consideration, we feel that it has merit but does not fully meet PLOS ONE’s publication criteria as it currently stands. Therefore, we invite you to submit a revised version of the manuscript that addresses the points raised during the review process.

Both reviewers agree that the current manuscript has merit, but have raised some questions about the design, analysis, and interpretation. Please address each of these thoroughly either by editing the manuscript and approach or rebutting reviewer points in a response. Note that reviewers will need to be satisfied with the methodological details (e.g., statistical analyses) and their interpretation before we can accept the manuscript, however I expect that it will be possible to do this sufficiently and thus am recommending a major revision.

We look forward to receiving your revised manuscript.

Kind regards,

Tyler Davis, Ph.D.

Academic Editor

PLOS ONE

Journal Requirements:

3. Thank you for including your ethics statement:  The study had local ethics committee approval (2122) and was conducted in accordance with APA ethics codes.

Please provide additional details regarding participant consent. In the ethics statement in the Methods and online submission information, please ensure that you have specified (1) whether consent was informed and (2) what type you obtained (for instance, written or verbal, and if verbal, how it was documented and witnessed). If your study included minors, state whether you obtained consent from parents or guardians. If the need for consent was waived by the ethics committee, please include this information.

Additional Editor Comments (if provided):

Reviewers' comments:

Reviewer's Responses to Questions

**Comments to the Author**

1. Is the manuscript technically sound, and do the data support the conclusions?

Reviewer #1: No

Reviewer #2: Yes

2. Has the statistical analysis been performed appropriately and rigorously? 

Reviewer #1: No

Reviewer #2: No

3. Have the authors made all data underlying the findings in their manuscript fully available?

Reviewer #1: Yes

Reviewer #2: Yes

4. Is the manuscript presented in an intelligible fashion and written in standard English?

Reviewer #1: Yes

Reviewer #2: Yes

5. Review Comments to the Author

Reviewer #1: The current paper presents the changes in the sustained posterior negativity (SPN) in response to the repetition of reflection/regularity stimuli. The authors claim that "SPN priming transferred between vertical and horizontal axis orientations (Experiment 4, N=48) and between reflectional and rotational symmetry". Large amount of work has been put in the manuscript and a number of experiments have been performed. Unfortunately, I have my concerns regarding the performed analysis and the interpretation of the results.

- No mentioning of a baselining of the data. Please, clarify if it was done and what was the window

- The analysis of three separate windows (first, second and third presentation) especially in the figures, make it difficult to compare the EEG responses to these stimuli. A better definition of SPN and its highlighting on the on the figures would be helpful.

- Part of the statistical analysis was done with rmANOVA but the degrees of freedom (DF) for it do not always match the number of observations or conducted comparisons. For instance, for mrANOVA, the following cannot be true: F(1.520,

71.437)=.... As far as I am aware, the DFs for rmANOVA are always an integer. Please, check it again and correct if necessary.

- The significant interactions require further multiple comparisons with correction of p-values. It would help to better interpret the results.

- I don't think using only 64 channel EEG for source localization is enough for making sound claims about the location of the effect (especially the ones that can go into the title). I would be more careful with the interpretation of the source localization data.

- In the experiment 1 it is mentioned that the data from grand-average of posterior electrodes was chosen, why not use the entire scalp and separate it in ROIs. That way one can see better a "spatial" distribution and development of the effect.

- Was a speeded response or a delayed response used for the task?

- The authors claim that they used an implicit task to avoid the repetition effect, but this effect can be observed without an explicit task and especially in the later experiments, the spatters (the random images) were starting to resemble faces. Since our cognition is prone to recognizing faces everywhere, how would authors deal with the issue of changes in N170 (an ERP negativity responsive to face processing) that temporally should overlaps with SPN.

- In the same vein, did the authors asked their participants if they recognized some pictures in the random patterns? The behavioral response would exclude the possibility of subjects giving a semantic background to their stimuli and therefore would avoid other type of priming.

- Why the oddball results were not included in the study? It would have been interesting to see if the priming effect stands after the interruption with the middle blank stimulus.

- Especially that the behavioral results were quite high, why not exclude the EEG trials with wrong behavioral results. This way, the data would have been better controlled.

- In the discussion on experiment 1, the authors claim: "It seems that there is something special about the onset of new reflection exemplars, which increases activation of the extrastriate SPN generators." Have you thought of a surprise effect? Breaking of a regular pattern can lead to the change in EEG-response around 250 ms post-onset (mismatch negativity) which would also co-occur with SPN.

- The source localization part on experiment 2 should go into supplementary materials since it does not provide any statistically relevant results.

- Please refer to all the results with p-value above 0.05 as statistically non-significant.

- Please specify the total number of ICAs out of which you mentioned how many you removed.

- The authors claim: "we have previously found that large alterations to band pass filters, ICA or trial exclusion rules do not change the shape of SPN wave dramatically". I would do this claim since even though the shape of SPN might not change, its variability might change drastically due to considerably decreased SNR. This would very much influence your results on statistical analysis.

- Please change the figures with ERP presentation so that the comparison between the SPNs are more clear and that the reader can better distinguish between the ERP components.

Reviewer #2: In this paper, Makin et al. measured the ERP for a sequence of symmetric or random patterns. They studied how the SPN changed with various stimulus parameters. The result can help to establish an ERP paradigm that discerns whether two stimulus parameters are processed by the same symmetry processing mechanisms. In general, this paper is well written and the result is useful. However, some issues need to be addressed before publication.

Statistics. The use of post-hoc statistics was not problematic. For instance, in Exp 1 (P. 8), the authors first performed a three-way ANOVA, followed by a two-way ANOVA, and then a one-way ANOVA on the third SPN and a t-test. There are several issues here. The first three-way ANOVA is obviously the omnibus ANOVA for the whole experiment and thus all the subsequent analyses should be considered as its post-hoc analyses. Since the authors found significant effects from sequence type, sequence position, and their interaction, the next step should be testing the sequence type effect at different sequence positions or vice versa. So, the one-way ANOVA and the t-test at the end are in the right direction, but the two-ANOVA is out-of-place. It provides no extra information from the three-way ANOVA and was misleading and thus should be removed. Even the one-way ANOVA and the t-test have their own issues. As post-hoc analyses, their error term should be the same as the omnibus ANOVA. Thus, why the degree-of-freedom for MSE of the ANOVAs and the t-test in these tests are different? Furthermore, the correction of significance level for multiple comparisons was not reported. Similar issues also exist for other experiments.

P. 11 First paragraph. Here, the authors rejected two alternative explanations of their data. However, these should be elaborated. For instance, it is not obvious to me why “If the priming effect was inherited from lower visual areas (Feuerriegel, 2016), it would have been [sic] stronger in the identical reflections condition”. The weaker response for the identical reflections condition might come from adaptation and has nothing to do with whether the effect is from the lower visual areas or not.

6. PLOS authors have the option to publish the peer review history of their article (what does this mean?). If published, this will include your full peer review and any attached files.

Reviewer #1: No

Reviewer #2: No

---

## [Author Response · Author response to Decision Letter 0]

28 May 2021

Please find the 20 page response to reviewers and editor comments in the attachments. 

Best wishes

Alexis Makin

---

## [Decision Letter · Decision Letter 1]

25 Jun 2021

Electrophysiological priming effects demonstrate independence and overlap of visual regularity representations in the extrastriate cortex.

PONE-D-20-34598R1

Dear Dr. Makin,

We’re pleased to inform you that your manuscript has been judged scientifically suitable for publication and will be formally accepted for publication once it meets all outstanding technical requirements.

Kind regards,

Tyler Davis, Ph.D.

Academic Editor

PLOS ONE

Additional Editor Comments (optional):

Reviewers' comments:

Reviewer's Responses to Questions

**Comments to the Author**

1. If the authors have adequately addressed your comments raised in a previous round of review and you feel that this manuscript is now acceptable for publication, you may indicate that here to bypass the “Comments to the Author” section, enter your conflict of interest statement in the “Confidential to Editor” section, and submit your "Accept" recommendation.

Reviewer #2: All comments have been addressed

2. Is the manuscript technically sound, and do the data support the conclusions?

Reviewer #2: Yes

3. Has the statistical analysis been performed appropriately and rigorously? 

Reviewer #2: Yes

4. Have the authors made all data underlying the findings in their manuscript fully available?

Reviewer #2: Yes

5. Is the manuscript presented in an intelligible fashion and written in standard English?

Reviewer #2: No

6. Review Comments to the Author

Reviewer #2: (No Response)

7. PLOS authors have the option to publish the peer review history of their article (what does this mean?). If published, this will include your full peer review and any attached files.

Reviewer #2: No

---

## [Editor Report · Acceptance letter]

30 Jun 2021

PONE-D-20-34598R1 

Electrophysiological priming effects demonstrate independence and overlap of visual regularity representations in the extrastriate cortex. 

Dear Dr. Makin:

I'm pleased to inform you that your manuscript has been deemed suitable for publication in PLOS ONE. Congratulations! Your manuscript is now with our production department. 

Kind regards, 

on behalf of

Dr. Tyler Davis 

Academic Editor

PLOS ONE